# The challenges of containing SARS-CoV-2 via test-trace-and-isolate

Sebastian Contreras [1,2,5], Jonas Dehning[1,5], Matthias Loidolt [1,5], Johannes Zierenberg [1], F. Paul Spitzner[1], Jorge H. Urrea-Quintero[1], Sebastian B. Mohr [1], Michael Wilczek [1,3], Michael Wibral[4] & Viola Priesemann [1,3]✉

Without a cure, vaccine, or proven long-term immunity against SARS-CoV-2, test-trace-and-isolate (TTI) strategies present a promising tool to contain its spread. For any TTI strategy, however, mitigation is challenged by pre- and asymptomatic transmission, TTI-avoiders, and undetected spreaders, which strongly contribute to "hidden" infection chains. Here, we study a semi-analytical model and identify two tipping points between controlled and uncontrolled spread: (1) the behavior-driven reproduction number $R_t^H$ of the hidden chains becomes too large to be compensated by the TTI capabilities, and (2) the number of new infections exceeds the tracing capacity. Both trigger a self-accelerating spread. We investigate how these tipping points depend on challenges like limited cooperation, missing contacts, and imperfect isolation. Our results suggest that TTI alone is insufficient to contain an otherwise unhindered spread of SARS-CoV-2, implying that complementary measures like social distancing and improved hygiene remain necessary.

[1] Max Planck Institute for Dynamics and Self-Organization, Am Faßberg 17, 37077 Göttingen, Germany. [2] Centre for Biotechnology and Bioengineering, Universidad de Chile, Beauchef 851, 8370456 Santiago, Chile. [3] Institute for the Dynamics of Complex Systems, University of Göttingen, Friedrich-Hund-Platz 1, 37077 Göttingen, Germany. [4] Campus Institute for Dynamics of Biological Networks, University of Göttingen, Hermann-Rein-Straße 3, 37075 Göttingen, Germany. [5] These authors contributed equally: Sebastian Contreras, Jonas Dehning, Matthias Loidolt. ✉email: viola.priesemann@ds.mpg.de

After SARS-CoV-2 started spreading rapidly around the globe in early 2020, many countries have successfully curbed the initial exponential rise in case of numbers ("first wave"). Most of the successful countries employed a mix of measures combining hygiene regulations and mandatory physical distancing to reduce the reproduction number and the number of new infections[1,2] together with testing, contact tracing, and isolation (TTI) of known cases[3,4]. Among these measures, those aimed at distancing—like school closures and a ban of all unnecessary social contacts ("strict lockdown")—were highly controversial, but have proven effective[1,2]. Notwithstanding, distancing measures put an enormous burden on society and the economy. In countries that have controlled the initial outbreak, there is a strong motivation to relax distancing measures, albeit under the constraint to keep the spread of COVID-19 under control[5,6].

In principle, it seems possible that both goals can be reached when relying on the increased testing capacity for SARS-CoV-2 infections if complemented by contact tracing and quarantine measures (e.g., like TTI strategies[4]); South Korea and Singapore illustrate the success of such a strategy[7–9]. In practice, resources for testing are still limited and costly, and health systems have capacity limits for the number of contacts that can be traced and isolated; these resources have to be allocated wisely to control disease spread[10].

TTI strategies have to overcome several challenges to be effective. Infected individuals can become infectious before developing symptoms[11,12], and because the virus is quite infectious, it is crucial to minimize testing-and-tracing delays[13]. Furthermore, SARS-CoV-2 infections generally appear throughout the whole population (not only in regional clusters), which hinders an efficient and quick implementation of TTI strategies.

Hence, these challenges that impact and potentially limit the effectiveness of TTI need to be incorporated together into one model of COVID-19 control, namely (1) the existence of asymptomatic, yet infectious carriers[14,15]—which are a challenge for symptom-driven but not for random-testing strategies; (2) the existence of a certain fraction of the population that is opposed to taking a test, even if symptomatic[16]; (3) the capacity limits of contact tracing and additional imperfections due to imperfect memory or non-cooperation of the infected. Last, enormous efforts are required to completely prevent the influx of COVID-19 cases into a given community, especially during the current global pandemic situation combined with relaxed travel restrictions[5,17]. This influx makes virus eradication impossible; it only leaves a stable level of new infections or their uncontrolled growth as the two possible regimes of disease dynamics. Thus, policymakers at all levels, from nations to federal states, all the way down to small units like enterprises, universities, or schools, are faced with the question of how to relax physical distancing measures while confining COVID-19 progression with the available testing and contact-tracing capacity[18].

Here, we employ a compartmental model of SARS-CoV-2 spreading dynamics that incorporates the challenges (1)–3). We base the model parameters on literature or reports using the example of Germany. The aim is to determine the critical value for the reproduction number in the general (not quarantined) population ($R_{crit}^H$), for which disease spread can still be contained. We find that—even under optimal use of the available testing and contract tracing capacity—the "hidden" reproduction number $R_t^H$ has to be maintained at sufficiently low levels, namely $R_t^H < R_{crit}^H \approx$ 2 (95% CI: 1.42–2.70). Hence, hygiene and physical distancing measures are required in addition to TTI to keep the virus spread under control. To further assist the efficient use of resources, we investigate the relative merits of contact tracing, symptom-driven testing, and random testing. We demonstrate the danger of a

tipping point associated with the limited capacity of tracing contacts of infected people. Finally, we show how either testing scheme has to be increased to re-stabilize disease spread after an increase in the reproduction number.

## Results

**Model overview**. We developed a SIR-type model[19,20] with multiple compartments that incorporates the effects of test-trace-and-isolate (TTI) strategies (for a graphical representation of the model see Fig. 1 and Supplementary Fig. 1). We explore how TTI can contain the spread of SARS-CoV-2 for realistic scenarios based on the TTI system in Germany. A major difficulty in controlling the spread of SARS-CoV-2 are the cases that remain hidden and behave as the general population does, potentially having many contacts. We explicitly incorporate such a "hidden" pool $H$ into our model and characterize the spread within by the reproduction number $R_t^H$, which reflects the population's contact behavior. Cases remain hidden until they enter a "traced" pool through testing or by contact tracing of an individual that has already been tested positive (see Fig. 1). All individuals in the traced pool $T$ isolate themselves (quarantine), reducing the reproduction number to $R_t^T$. Apart from a small leak, novel infections therein are then assumed to remain within the traced pool. We investigate both symptom-driven and random testing, which differ in the clinical characteristics of the cases they can reveal: random testing can, in principle, uncover even asymptomatic cases, while symptom-driven testing is limited to symptomatic cases willing to be tested. Parameters describing the spreading dynamics (Table 1) are based on the available literature on COVID-19[15,16,21–23], while parameters describing the TTI system are inspired by our example case of Germany wherever possible.

We provide the code of the different analyses at https://github.com/Priesemann-Group/covid19_tti (https://doi.org/10.5281/zenodo.4290679). An interactive platform to simulate scenarios different from those presented here is available on the same GitHub repository.

**TTI strategies can in principle control SARS-CoV-2 spread**. To demonstrate that TTI strategies can, in principle, control the disease spread, we simulated a new outbreak starting in the hidden pool (Fig. 2). We assume that the outbreak is unnoticed initially, and then evaluate the effects of two alternative testing and contact tracing strategies starting at day 0: Contact tracing is either efficient, i.e., 66% ($\eta = 0.66$) of the contacts of a positively tested person are traced and isolated without delay ("efficient tracing"), or contact tracing is assumed to be less efficient, identifying only 33% of the contacts ("inefficient tracing"). In both regimes, the default parameters are used (Table 1), which include symptom-driven testing with rate $\lambda_s = 0.1$, and isolation of all tested positively, which reduces their reproduction number by a factor of $\nu = 0.1$.

An efficient contact tracing rapidly depletes the hidden pool $H$ and populates the traced pool $T$, and thus stabilizes the total number of infections $T + H$ (Fig. 2a). The system relaxes to its equilibrium, which is a function of TTI and epidemiological parameters (Supplementary Eqs. (3)–(5)). Consequently, the observed number of daily infections ($\hat{N}^{obs}$) approaches a constant value (Fig. 2b), while the observed reproduction number $\hat{R}_t^{obs}$ approaches unity (Fig. 2c), further showing that effective TTI can be sufficient to stabilize the disease spread with $R_t^H = 1.8$.

In contrast, inefficient contact tracing cannot deplete the hidden pool sufficiently quickly to stabilize the total number of infections (Fig. 2d). Thus, the absolute and the observed daily

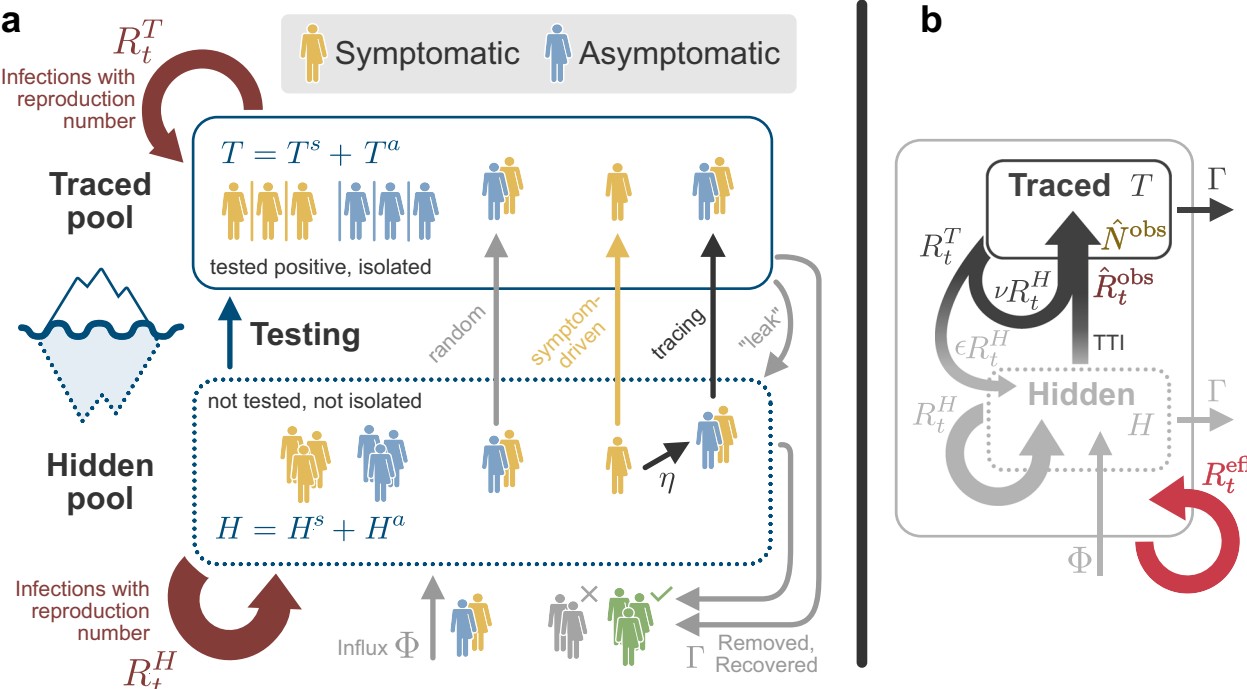

**Fig. 1 Illustration of interactions between the hidden $H$ and traced $T$ pools in our model. a** In our model, we distinguish two different infected population groups: the one that contains the infected individuals that remain undetected until tested (hidden pool $H$), and the one with infected individuals that we already follow and isolate (traced pool $T$). Super indexes $s$ and $a$ in both variables account for symptomatic and asymptomatic individuals. Until noticed, an outbreak will fully occur in the hidden pool, where case numbers increase according to this pool's reproduction number $R_t^H$. Testing and tracing of hidden infections transfers them to the traced pool and helps to empty the hidden pool; this prevents offspring infections and reduces the overall growth of the outbreak. Due to the self-isolation imposed in the traced pool, its reproduction number $R_t^T$ is expected to be considerably smaller than $R_t^H$, and typically smaller than 1. Once an individual is tested positive, all the contacts since the infection are traced with some efficiency ($\eta$). Two external events further increase the number of infections in the hidden pool, namely, the new contagions occurring in the traced pool that leak to the hidden pool and an influx of externally acquired infections ($\Phi$). In the absence of new infections, pool sizes are naturally reduced due to recovery (or removal), proportional to the recovery rate $\Gamma$. **b** Simplified depiction of the model showing the interactions of the two pools. New infections generated in the traced pool can remain there ($\nu$) or leak to the hidden pool ($\epsilon$). Note that the central epidemiological observables are highlighted in color: The $\hat{N}^{\mathrm{obs}}$ (brown) and $\hat{R}_t^{\mathrm{obs}}$ (dark red) can be inferred from the traced pool, but the effective reproduction number $\hat{R}_t^{\mathrm{eff}}$ (light red) that governs the stability of the whole system remains hidden.

**Table 1 Model parameters.**

| Parameter | Meaning | Value (default) | Range | Units | Source |
|---|---|---|---|---|---|
| $M$ | Population size | 80,000,000 | | People | Assumed |
| $R_t^H$ | Reproduction number (hidden) | 1.80 | | – | 2,67,68 |
| $\Gamma$ | Recovery rate | 0.10 | 0.08–0.12 | Day$^{-1}$ | 58,69,70 |
| $\xi$ | Asymptomatic ratio | 0.15 | 0.12–0.33 | – | 22,23 |
| $\varphi$ | Fraction skipping testing | 0.20 | 0.10–0.40 | – | 16 |
| $\nu$ | Isolation factor (traced) | 0.10 | | – | Assumed |
| $\lambda_r$ | random-testing rate | 0 | 0–0.02 | Day$^{-1}$ | Assumed |
| $\lambda_s$ | symptom-driven testing rate | 0.10 | 0–1 | Day$^{-1}$ | Assumed |
| $\eta$ | Tracing efficiency | 0.66 | | – | Assumed |
| $N_{\max}$ | Maximal tracing capacity | ≈ 718 | 200–6000 | Cases day$^{-1}$ | Assumed[a] |
| $\epsilon$ | Missed contacts (traced) | 0.10 | | – | Assumed |
| $\Phi$ | Influx rate (hidden) | 15 | | Cases day$^{-1}$ | Assumed[a] |
| $\lambda_{r,\max}$ | Maximal test capacity per capita | 0.002 | | Cases day$^{-1}$ | 56,57 |
| $R_t^T$ | Reproduction number (traced) | 0.36 | | – | $R_t^T = (\nu + \epsilon)R_t^H$ |
| $\xi^{\mathrm{ap}}$ | Apparent asymptomatic ratio | 0.32 | | – | $\xi^{\mathrm{ap}} = \xi + (1-\xi)\varphi$ |
| $R_{\mathrm{crit}}^H$ | Critical reproduction number (hidden) | 1.89 | | – | Numerically calculated from model parameters |

[a]Chosen for a country with a population of $M = 80 \cdot 10^6$. See "Methods" for considerations.

number of infections $N$ continue to grow approximately exponentially (Fig. 2e). In this case, the TTI strategy with ineffective contact tracing slows the spread but cannot control the outbreak.

**TTI extends the stabilized regimes of spreading dynamics.** Comparing the two TTI strategies from above demonstrates that two distinct regimes of spreading dynamics are attainable under the condition of a nonzero influx of externally acquired infections

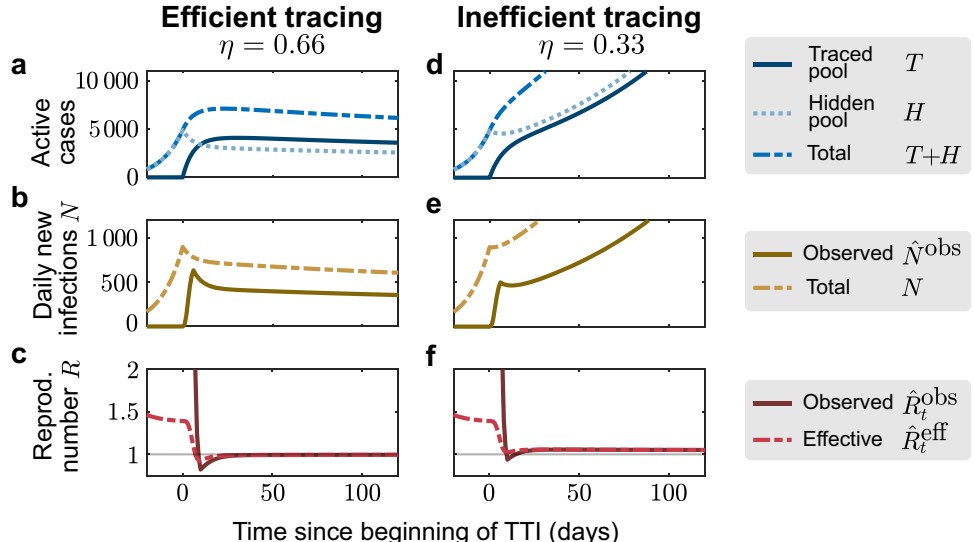

**Fig. 2 Sufficient testing and contact tracing can control the disease spread, while insufficient TTI only slows it.** We consider a test-trace-and-isolate (TTI) strategy with symptom-driven testing ($\lambda_s = 0.1$) and two tracing scenarios: For high tracing efficiency ($\eta = 0.66$, **a–c**), the outbreak can be controlled by TTI; for low tracing efficiency ($\eta = 0.33$, **d, e**) the outbreak cannot be controlled because tracing is not efficient enough. **a, d** The number of infections in the hidden pool grows until the outbreak is noticed on day 0, at which point symptom-driven testing ($\lambda_s = 0.1$) and contact tracing ($\eta$) starts. **b, e** The absolute number of daily infections ($N$) grows until the outbreak is noticed on day 0; the observed number of daily infections ($\hat{N}^{\text{obs}}$) shown here is simulated as being inferred from the traced pool and subject to a gamma-distributed reporting delay with a median of 4 days. **c, f** The observed reproduction number ($\hat{R}_t^{\text{obs}}$) is estimated from the observed new infections ($\hat{N}^{\text{obs}}$), while the effective reproductive number ($\hat{R}_t^{\text{eff}}$) is estimated from the total daily new infections ($N$). After an initial growth period, it settles to $\hat{R}_t^{\text{obs}} = 1$ if the outbreak is controlled (efficient tracing), or to $\hat{R}_t^{\text{obs}} > 1$ if the outbreak continues to spread (inefficient tracing). All the curves plotted are obtained from numerical integration of Eqs. (1)–(5).

$\Phi$: The system either evolves towards some intermediate but stable number of new cases $N$ (Fig. 2a–c), or it is unstable, showing a steep growth (Fig. 2d–f). These two dynamical regimes are characterized—after an initial transient—by different "observed" reproduction numbers $\hat{R}_t^{\text{obs}}$, inferred from the new cases of the traced pool $\hat{N}^{\text{obs}}$. If $\hat{R}_t^{\text{obs}} < 1$, the outbreak is under control (solid line in Fig. 2c), while for $\hat{R}_t^{\text{obs}} > 1$ the outbreak continues to spread (Fig. 2f). The former regime extends the "stable" regime of the simple SIR model beyond $R_t^H = 1$ and thus constitutes a novel "TTI-stabilized" regime of spreading dynamics (see below, and Supplementary Fig. 5 for the full phase diagram).

**Limited TTI requires a safety margin to maintain stability.** Having demonstrated that an effective TTI strategy can, in principle, control the disease spread, we now turn towards the problem of limited TTI capacity. So far, we assumed that the efficiency of the TTI strategy does not depend on the absolute number of cases. Yet, the amount of contacts that can reliably be traced by health authorities is limited due to the work to be performed by trained personnel: Contact persons have to be identified, informed, and ideally also counseled during the preventive quarantine. Exceeding this limit causes delays in the process, which will eventually become longer than the generation time of 4 days—rendering contact tracing ineffective. We model this tracing capacity as a hard cap $N_{\text{max}}$ on the number of contacts that can be traced each day and explore its effects on stability.

As an example of how this limited tracing capacity can cause a new tipping point to instability, we simulate here a short but large influx of externally acquired infections (a total of 4000 hidden cases with 92% occurring in the 7 days around $t = 0$, normally distributed with $\sigma = 2$ days, see Fig. 3). This exemplary influx aims to resemble the large number of German holidaymakers returning from summer vacation. It is a rather conservative estimate given that there were 900 such cases observed in the first

two weeks of July at Bavarian highway test-centers alone[24]. We set two different tracing-capacity limits, reached when the observed number of daily new cases $\hat{N}^{\text{obs}}$ reaches $N_{\text{max}} = 718$ (or $N_{\text{max}} = 470$) observed cases per day (see "Methods"). In both scenarios, the sudden influx leads to a jump of infections in the hidden pool (Fig. 3a, d), followed by a rapid increase in new traced cases (Fig. 3b, e). With sufficiently high tracing capacity, the outbreak can then be contained, because during the initial shock $\hat{N}^{\text{obs}}$ does not exceed the capacity limit $N_{\text{max}}$ (Fig. 3b, brown vs gray lines). In contrast, with lower capacity, the outbreak accelerates as soon as the observed new cases $\hat{N}^{\text{obs}}$ exceeds the capacity limit $N_{\text{max}}$. Not only the capacity limit but also the amplitude of the influx (Supplementary Fig. 3), its duration (Supplementary Fig. 4), or whether it occurs periodically (Fig. 4) can decide whether the observed new cases $\hat{N}^{\text{obs}}$ exceed the capacity limit $N_{\text{max}}$ and cause a tipping-over into instability. In particular, periodic influxes (e.g., holidays) may cause the tipping-over not necessarily because of a single event but due to their cumulative impact. These scenarios demonstrate that the limited tracing capacity renders the system metastable. If the capacity limit is exceeded due to some external perturbation, the tracing cannot compensate the perturbation, and the spread gets out of control.

Even without a large influx event, the tipping-over into instability can occur when a relaxation of contact restrictions causes slow growth in case numbers. This slow growth will accelerate dramatically once the tracing capacity limit is reached —constituting a transition from a slightly unstable to a strongly unstable regime (Fig. 5 and Supplementary Fig. 5d). To illustrate this, we simulated an increase of the hidden reproduction number $R_t^H$ (of a system in stable equilibrium) at $t = 0$, from the subcritical default value of $R_t^H = 1.8$ to a supercritical value $R_t^H = 2$, which renders the system slightly unstable (Fig. 6). At $t = 0$, the case numbers start to grow slowly until the observed number of new cases exceeds the tracing capacity limit $N_{\text{max}}$.

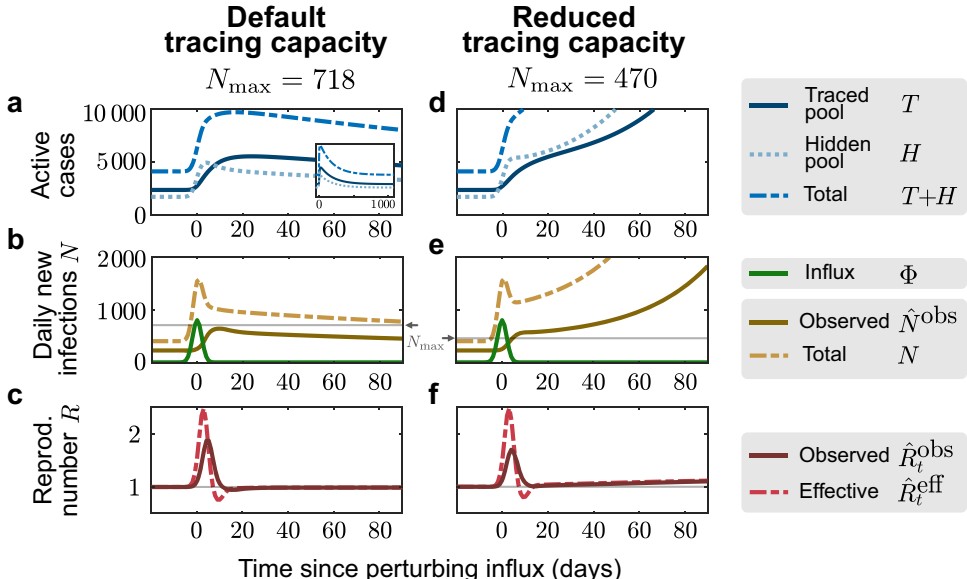

**Fig. 3 Finite tracing capacity makes the system vulnerable to large influx events.** A single large influx event (a total of 4000 hidden cases with 92% occurring in the 7 days around $t=0$, normally distributed with standard deviation $\sigma = 2$ days) drives a metastable system with reduced tracing capacity (reached at $N_{max} = 470$) to a new outbreak (**d–f**), whereas a metastable system with our default tracing capacity (reached at $N_{max} = 718$) can compensate a sudden influx of this size (**a–c**). **a, d** The number of infections in the hidden pool (dotted) jump due to the influx event at $t=0$, and return to stability for default capacity (**a**) or continue to grow in the system with reduced capacity (**d**). Correspondingly, the number of cases in the traced pool (solid line) either slowly increases after the event and absorbs most infections before returning to stability (inset in **a**, time axis prolonged to 1000 days), or proceeds to grow steeply (**d**). **b, e** The absolute number of new infections (dashed, yellow) jumps due to the large influx event (solid green line). The number of daily observed cases (solid brown line) slowly increases after the event, and relaxes back to baseline (**a**), or increases fast upon exceeding the maximum number of new observed cases $N_{max}$ (solid gray line) for which tracing is effective. **c, f** The effective (dashed red line) and observed (solid dark red line) reproduction numbers change transiently due to the influx event before returning to 1 for the default tracing capacity. In the case of a reduced tracing capacity and a new outbreak, they slowly begin to grow afterward (**f**). All the curves plotted are obtained from numerical integration of Eqs. (1)–(5).

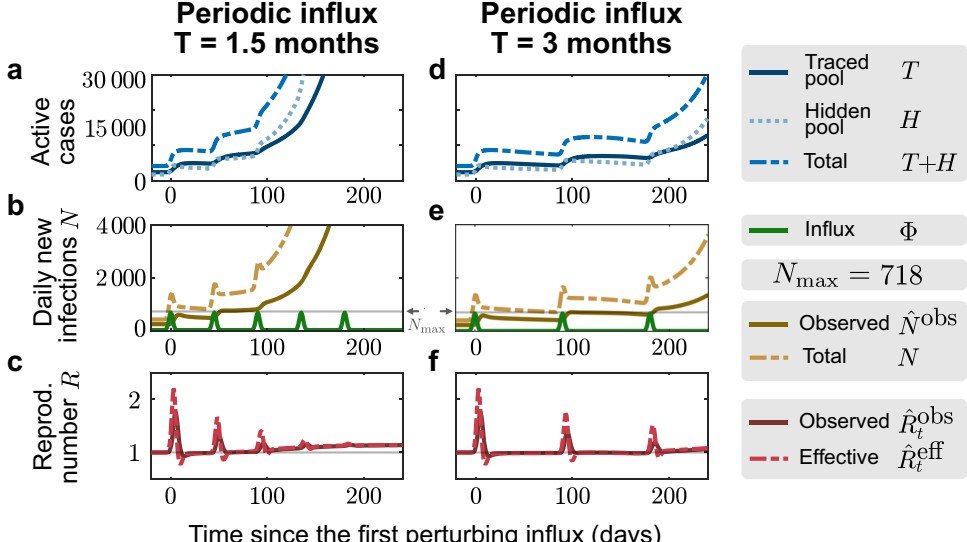

**Fig. 4 Manageable influx events that recur periodically can overwhelm the tracing capacity.** For the default capacity scenario, we explore whether periodic influx events can overwhelm the tracing capacity: A 'manageable" influx that would not overwhelm the tracing capacity on its own (3331 externally acquired infections, 92% of which occur in 7 days) repeats every 1.5 months (**a–c**) or every 3 months (**d–f**). In the first case, the system is already unstable after the second event because case numbers remained high after the first influx (**b**). In the second case, the system remains stable after both the first and second event (**e**), but it becomes unstable after the third (**f**).

From thereon, the tracing system breaks down, and the growth self-accelerates. This is reflected in the steep rise of new cases after day 100—thus with a considerable delay after the change of $R_t^H$, i.e., the population's behavior.

Both the initial change in the hidden reproduction number and the breakdown of the tracing system are reflected in the observed reproduction number $\hat{R}_t^{obs}$ (Fig. 6c). It transits from stability ($\hat{R}_t^{obs} = 1$) to instability ($\hat{R}_t^{obs} > 1$). However, the absolute values

of $\hat{R}_t^{\text{obs}}$ are not very indicative of the public's behavior ($R_t^H$), because already small changes in $R_t^H$ can induce large transient changes in $\hat{R}_t^{\text{obs}}$. In our example, $\hat{R}_t^{\text{obs}}$ shows a strong deflection after $t = 0$, although $R_t^H$ changes only slightly; later, at $t \approx 100$ it starts to ramp to a new value, although $R_t^H$ did not change. This ramping is due to the tracing capacity $N_{\text{max}}$ being exceeded, which accelerates the spread. $\hat{R}_t^{\text{obs}}$ finally approaches a new steady-state value, as sketched in Supplementary Fig. 5d. To summarize, deducing the stability of the spread from $\hat{R}_t^{\text{obs}}$ is

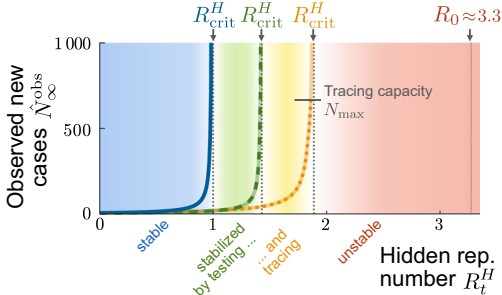

**Fig. 5 Testing and tracing give rise to two TTI-stabilized regimes of spreading dynamics.** In addition to the intrinsically stable regime of the simple SIR model (blue region), our model exhibits two TTI-stabilized regimes that arise from the isolation of formerly "hidden" infected individuals uncovered through symptom-based testing alone (green region) or additional contact tracing (amber region). Due to the external influx, the number of observed new cases reaches a nonzero equilibrium $\hat{N}_\infty^{\text{obs}}$ that depends on the hidden reproductive number (colored lines). These equilibrium numbers of new cases diverge when approaching the respective critical hidden reproductive numbers ($R_{\text{crit}}^H$) calculated from linear stability analysis (dotted horizontal lines). Taking into account a finite tracing capacity $N_{\text{max}}$ shrinks the testing-and-tracing stabilized regime and makes it metastable (dotted amber line). Note that, for our standard parameter set, the natural base reproduction number $R_O$ lies in the unstable regime. Please see Supplementary Fig. 5 for a full phase diagram and Supplementary Note 1 for the linear stability analysis.

challenging because $\hat{R}_t^{\text{obs}}$ reacts very sensitively to many types of transients. $R_t^H$, in contrast, would be a reliable indicator of true spreading behavior but is not accessible easily.

**Imperfect TTI would require further containment measures.** Above, we illustrated that a combination of symptom-driven testing and contact tracing could control the outbreak for a default reproduction number of $R_t^H = 1.8$. We now ask how efficient the TTI scheme and implementation must be to control the disease for a range of reproduction numbers—i.e., what TTI parameters are necessary to avoid the tipping over to $\hat{R}_t^{\text{eff}} > 1$. To this end, we perform linear stability analysis to calculate the critical reproduction number at which the tipping-over occurs (see Supplementary Eq. (1) in Supplementary Note 1). When assessing stability not only for a single scenario along the $R_t^H$-axis but for multiple parameter combinations, the tipping points turn into critical lines (or surfaces). Here, we examine how these critical lines depend on different combinations of symptom-driven testing, random testing, and contact tracing.

Random testing with tracing, but without symptom-driven testing ($\lambda_s = 0$), is not sufficient to contain an outbreak (under our default parameters and $R_t^H \leq 1.5$; Fig. 7a). This is because the rate of random testing $\lambda_r$ would have to be unrealistically large. It exceeds the current capacity of testing ($\lambda_{r,\text{max}} \sim 0.002$, see "Methods" for details), even if ten tests are pooled ($\lambda_r \sim 10\lambda_{r,\text{max}}$[25]). Thus, the contribution of symptom-driven testing is necessary to control any realistic new outbreak through TTI.

Contact tracing markedly contributes to outbreak mitigation (Fig. 7b). In its absence, i.e., when isolating only individuals that were positive in a symptom-driven or random test, the outbreak can be controlled for intermediate reproduction numbers ($R_t^H < 2.5$ in Fig. 7b) but not for higher ones if the limit of $\lambda_{r,\text{max}} < 0.02$ is respected.

The most effective combination appears to be symptom-driven testing together with contact tracing (Fig. 7c). This combination shows stability even for spreads close to the basic reproduction number $R_t^H = R_0 \approx 3.3$[21,26,27], when implemented extremely efficiently (e.g., with $\lambda_s = 0.66$ and $\eta = 0.66$). However, this

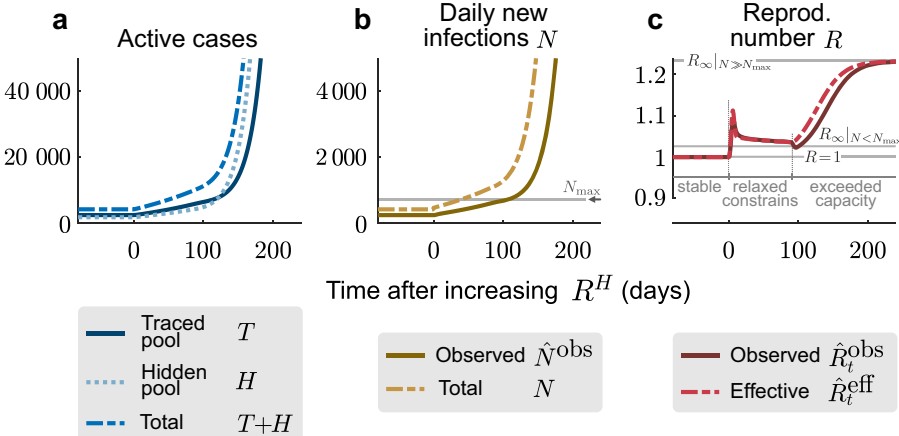

Traced pool $T$
Hidden pool $H$
Total $T+H$

Observed $\hat{N}^{\text{obs}}$
Total $N$

Observed $\hat{R}_t^{\text{obs}}$
Effective $\hat{R}_t^{\text{eff}}$

**Fig. 6 A relaxation of restrictions can slowly overwhelm the finite tracing capacity and trigger a new outbreak. a** At $t = 0$, the hidden reproduction number increases from $R_t^H = 1.8$ to $R_t^H = 2.0$ (i.e., slightly above its critical value). This leads to a slow increase in traced active cases (solid blue line). **b** When the number of observed new cases (solid brown line) exceeds the tracing capacity limit $N_{\text{max}}$ (solid gray line), the tracing system breaks down, and the outbreak starts to accelerate. **c** After an initial transient at the onset of the change in $R_t^H$, the observed reproduction number (solid red line) faithfully reflects both the slight increase of the hidden reproduction number due to relaxation of contact constraints, and the strong increase after the tracing capacity (solid gray line) is exceeded at $t \approx 100$. In both cases, the observed reproduction number $\hat{R}_t^{\text{obs}}$ approaches two different limit values $R_\infty$, which are derived from a linear stability analysis (further details in Supplementary Fig. 5). All the curves plotted are obtained from numerical integration of Eqs. (1)–(5).

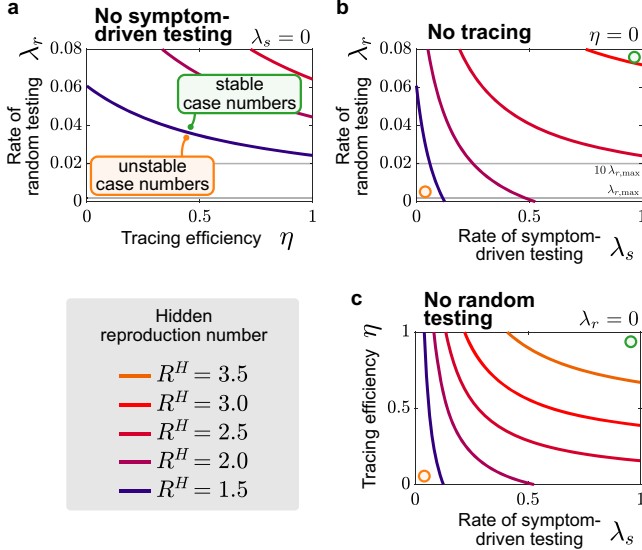

**Fig. 7 Symptom-driven testing and contact tracing need to be combined to control the disease.** Stability diagrams showing the boundaries (continuous curves) between the stable (controlled) and uncontrolled regimes for different testing strategies combining random testing (rate $\lambda_r$), symptom-driven testing (rate $\lambda_s$), and tracing (efficiency $\eta$). Gray lines in plots with $\lambda_r$-axes indicate capacity limits (for our example Germany) on random testing ($\lambda_{r,max}$) and when using pooling of ten samples, i.e., $10\lambda_{r,max}$. Colored lines depict the transitions between the stable and the unstable regime for a given reproduction number $R_t^H$ (color-coded). The transition from 'stable" to 'unstable" case numbers is explicitly annotated for $R_t^H = 1.5$ in panel **a**. **a** Combining tracing and random testing without symptom-driven testing is in all cases not sufficient to control outbreaks, as the necessary random tests exceed even the pooled testing capacity ($10\lambda_{r,max}$). **b** Combining random and symptom-driven testing strategies without any contract tracing requires unrealistically high levels of random testing to control outbreaks with large reproduction numbers in the hidden pool ($R_t^H > 2.0$). The required random tests to significantly change the stability boundaries exceed the available capacity in Germany $\lambda_{r,max}$. Even considering the possibility of pooling tests ($10\lambda_{r,max}$) often does not suffice to control outbreaks. **c** Combining symptom-driven testing and tracing suffices to control outbreaks with realistic testing rates $\lambda_s$ and tracing efficiencies $\eta$ for moderate values of reproduction numbers in the hidden pool, $R_t^H$, but fails to control the outbreak for large $R_t^H$. The curves showing the critical reproduction number are obtained from the linear stability analysis (Supplementary Eq. (1)).

implementation would require that all symptomatic persons get tested within 1–2 days after getting infectious, thus potentially already in their pre-symptomatic phase, which may be difficult to realize. (Note that the asymptomatic cases are already accounted for in the model and do not pose an additional problem). Considering these difficulties, the combination of symptom-driven testing and contact tracing appears to be sufficient to contain outbreaks with intermediate reproduction numbers ($R_t^H \sim 2$ can be controlled with e.g., $\lambda_s \leq 0.5$ and $\eta = 0.66$, Fig. 7c).

Overall, our model suggests that the combination of timely symptom-driven testing within very few days, together with isolation of positive cases and efficient contact tracing, can be sufficient to control the spread of SARS-CoV-2 given the reproduction number in the hidden pool is $R_t^H \approx 2$ or lower. For random testing at the population level to be effective, one would require much higher test rates than currently available in Germany. Nevertheless, random testing can be useful to control highly localized outbreaks and is paramount for screening frontline workers in healthcare, eldercare, and education.

**How can TTI allow the relaxation of contact constraints?** There are currently strong incentives to loosen restrictive measures and return to a more pre-COVID-19 lifestyle[28,29]. However, any such loosening can lead to a higher reproduction number $R_t^H$, which could potentially exceed the critical value $R_{crit}^H$, for which current TTI strategies ensure stability. To retain stability despite increasing $R_t^H$, this increase has to be compensated by stronger mitigation efforts, such as further improvement of TTI. Thereby the critical value $R_{crit}^H$ is effectively increased. In the following, we compare the capacity of the different TTI and model parameter changes to compensate for increases in the reproduction number $R_t^H$. In detail, we start from the highest reproduction number that can be controlled by the default parameters, $R_{crit}^H = 1.89$, and calculate how each model parameter would have to be changed to achieve the desired increase in $R_{crit}^H$. For all default parameters, see Table 1.

First, we explore how well an increase of random and symptom-driven test rates can compensate for an increase in $R_t^H$ (Fig. 8a). We find that population-wide random testing would need to increase extensively to compensate for increases in $R_t^H$, i.e., $\lambda_r$ quickly exceeds realistic values (gray lines in Fig. 8a). Thus, random testing at a whole population level is not the most efficient tool to compensate for increases of the hidden reproduction rate, but that does not diminish its usefulness in controlling localized outbreaks or protecting frontline workers and highly vulnerable populations.

In contrast, scaling up symptom-driven testing can in principle compensate an increase of $R_t^H$ up to about 3 (Fig. 8a). Beyond $R_t^H = 3$ and $\lambda_s \approx 0.4$, $\lambda_s$ increases more steeply, making this compensation increasingly costly (Fig. 8a). Furthermore, levels of $\lambda_s > 0.5$ seem hard to realize as they would require testing within $< 2$ days of becoming infectious, i.e., while many infected are still pre-symptomatic. Realistically, only moderate increases in $R_t^H$ can be compensated by decreasing the average delay of symptom-driven testing alone.

Tracing the contacts of an infected person and asking them to quarantine preventively is a vital contribution to contain the spread of SARS-CoV-2 if done without delay[3,13]. As a default value, we assumed that a fraction $\eta = 0.66$ of contacts are traced and isolated within a day. This fraction can, in principle, be increased further to compensate for an increase in $R_t^H$ and still guarantee stability (Fig. 8a). However, because $\eta$ is already high in the first place, its range is quite limited, and even perfect contact tracing cannot compensate for an $R_t^H$ of 2.5. More elaborate contact tracing strategies, like backward-forward tracing, might further improve its practical efficacy.

As an alternative to improved TTI rates and efficiencies, improved compliance may compensate for an increase in $R_t^H$: One might aim to reduce the number of contacts missed in the traced pool $\epsilon$, improve the isolation factor $\nu$, or reduce the fraction of people avoiding tests despite showing symptoms $\varphi$ (Fig. 8c). These improvements might be more challenging to achieve from a policymaker perspective but could be targeted by educational and awareness-raising campaigns. However, since we assumed already in the default scenario that the behavioral factors ($\epsilon$, $\nu$, $\varphi$) are not too large, the potential improvement is limited.

The amount of reduction achievable by each method is limited, which calls to leverage all these strategies together. Furthermore, as can be seen from the curvature of the lines in Fig. 7, the beneficial effects are synergistic, i.e., they are larger when combining several strategies instead of spending twice the efforts on a unique one. This synergy of improved TTI measures and awareness campaigning could relax contact constraints while keeping outbreaks under control. Nonetheless, our model still

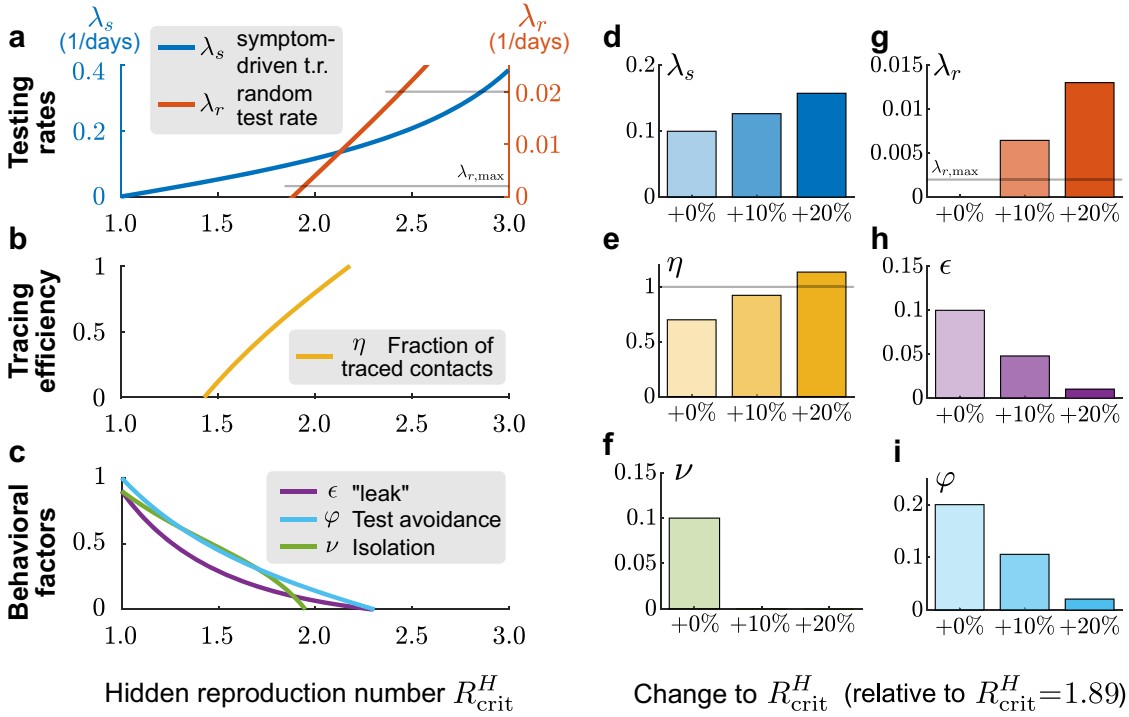

**Fig. 8 Adapting testing strategies allows the relaxation of contact constraints to some degree.** The relaxation of contact constraints increases the reproduction number of the hidden pool $R_t^H$, and thus needs to be compensated by adjusting model parameters to keep the system stable. **a–c** Value of a single parameter required to keep the system stable despite a change in the hidden reproduction number, while keeping all other parameters at default values. **a** Increasing the rate of symptom-driven testing ($\lambda_s$, blue) can in principle compensate for hidden reproduction numbers close to $R_0$. However, this is optimistic as it requires that anyone with symptoms compatible with COVID-19 gets tested and isolated on average within 2.5 days—requiring extensive resources and efficient organization. Increasing the random-testing rate ($\lambda_r$, red) to the capacity limit (for the example Germany, gray line $\lambda_{r,\max}$) would have almost no effect, pooling tests to achieve $10\lambda_{r,\max}$ can compensate partly for larger increases in $R_t^H$. **b** Increasing the tracing efficiency ($\eta$) can compensate only small increases in $R_t^H$. **c** Decreasing the fraction of symptomatic individuals who avoid testing ($\varphi$), the leak from the traced pool ($\epsilon$) or the escape rate from isolation ($\nu$) can in principle compensate for small increases in $R_t^H$. **d–i** To compensate a 10% or 20% increase of $R_t^H$, while still keeping the system stable, symptom-driven testing ($\lambda_s$) could be increased (**d**), or $\epsilon$ or $\varphi$ could be decreased (**h,i**). In contrast, only changing $\lambda_r$, $\eta$, or $\nu$ would not be sufficient to compensate a 10 % or 20 % increase in $R_t^H$, because the respective limits are reached (**e**, **f**, **g**). All parameter changes are computed through stability analysis (Supplementary Eq. (1)).

indicates that compensating the basic reproduction number $R_t^H = R_0 \approx 3.3$[21,26,27] might be very costly, and hence some degree of physical distancing might be required.

**Robustness against parameter changes and model limitations.** Above, we showed that changing the implementation of the TTI strategy can accommodate higher reproduction numbers $R_{\mathrm{crit}}^H$ — but how robust are these implementations against parameter uncertainties? To explore the robustness of the resulting hidden reproduction number $R_{\mathrm{crit}}^H$ against simultaneous variation of multiple TTI parameters, we draw these parameters from beta distributions (because all parameters are bounded by 0 and 1) centered on the default values, and perform an error propagation analysis (Supplementary Table 1). We found that a hidden reproduction number of $R_t^H \leq 1.4$ (95% CI, 1.23–1.69) can be compensated by testing alone, whereas additional contact tracing allows a hidden reproduction number of $R_t^H \leq 1.9$ (95% CI, 1.42–2.70, Supplementary Fig. 2 and Supplementary Table 1). This shows that the exact implementation of the TTI strategy strongly impacts the public behavior that can be controlled. However, none of them allows for a complete lifting of the contact restrictions ($R_0 = 3.3$).

However, not only the robustness against variation of parameters is an important aspect but also underlying

assumptions in the model structure. Our model also comes with some inevitable simplifications, but these do not compromise the conclusions drawn here. Specifically, our model is simple enough to allow for a mechanistic understanding of its dynamics and analytical treatment of the control and stability problems. This remains true even when extending the model to incorporate more biological realism, e.g., the different transmissibility of asymptomatic and symptomatic cases (Supplementary Fig. 6). Owing to its simplicity it has certain limitations: In contrast to agent-based simulations[30,31], we do not include realistic contact structures[4,5,32]—the infection probability is uniform across the whole population. This limitation will become relevant mostly when trying to devise even more efficient testing-and-tracing strategies or stabilizing a system very close to its tipping point. Compared to other mean-field based studies, which included a more realistic temporal evolution of infectiousness[33,34], we implicitly assume that infectiousness decays exponentially. This assumption has the disadvantage of making the interpretation of rate parameters more difficult, but should not affect the stability analyses presented here.

**Discussion**
Using a compartmental SIR-type model with realistic parameters based on our example case in Germany, we find that test-trace-and-isolate can, in principle, contain the spread of SARS-CoV-2 if

some physical distancing measures are continued. We analytically derived the existence of a novel metastable regime of spreading dynamics governed by the limited capacity of contact tracing and show how transient perturbations can tip a seemingly stable system into the unstable regime. Furthermore, we explored the boundaries of this regime for different TTI strategies and efficiencies of the TTI implementation.

Our results agree with other simulation and modeling studies investigating how efficient TTI strategies are in curbing the spread of the SARS-CoV-2. Both agent-based studies with realistic contact structures[4] and studies using mean-field spreading dynamics with tractable equations[33–37] agree that TTI measures are an important contribution to control the pandemic. Fast isolation is arguably the most crucial factor, which is included in our model in the testing rate $\lambda_s$. Yet, TTI is generally not perfect and the app-based solutions that have been proposed at present still lack the necessary large adoption that was initially foreseen, and that is necessary for these solutions to work[34]. Our work, as well as others[4,34,38,39], shows that realistic TTI can compensate reproduction numbers of around 1.5–2.5, which is however lower than the basic reproduction number of around 3.3[21,26,27]. This calls for continued contact reduction on the order of 25–55%, and it does highlight not only the importance of TTI but also the need for other mitigation measures.

Our work extends previous studies by combining the explicit modeling of a hidden pool (including test avoiders) to explore various ways of allocating testing-and-tracing resources. This allows us to investigate the effectiveness of multiple approaches to stabilize disease dynamics in the face of relaxation of physical distancing. This yields important insights for policymakers into how to allocate resources. We also include a capacity limit of tracing, which is typically not included in other studies. However, it is crucial to understand the metastable regime of a TTI-stabilized system and understand the importance of keeping a safety-distance to the critical reproduction number of a given TTI strategy. Last, we highlight the essential differences between the observed reproduction numbers—as they are reported in the media—and the more important, but hard to access, reproduction number in the hidden pool. Specifically, we show how the transient behavior of the observed reproduction number may be easily misinterpreted.

Limited TTI capacity implies a metastable regime with the risk of sudden explosive growth. Both testing and tracing contribute to containing the spread of SARS-CoV-2. However, if the number of new infections exceeds their capacity limit, an otherwise controlled spread becomes uncontrolled. This is particularly troubling because the spread is self-accelerating: the more the capacity limit is exceeded, the less testing and tracing can contribute to containment. The reproduction number has to stay below its critical value to avoid this situation and the number of new infections below TTI capacity. Therefore, it is advisable to maintain a safety margin to these limits. Otherwise, a small increase of the reproduction number, a super-spreading event[40], or a sudden influx of externally acquired infections e.g., after holidays, leads to uncontrolled spread. Re-establishing stability is then quite difficult.

As the number of available tests is limited, the relative efficiencies of random, symptom-driven and tracing-based testing should determine the allocation of resources[10]. The efficiency of test strategies in terms of the positivity rate is a primary metric to determine the allocation of tests[41]. Contact-tracing-based testing will generally be the most efficient use of tests (positivity rate on the order of $R_t^H / \{ \text{number of contacts} \}$), especially in the regime of low contact numbers[37,42]. The efficiency of symptoms-driven testing depends on the set of symptoms used for admission: Highly specific symptom sets will allow for a high yield, but miss a number of cases

(for instance, 33% of cases do not show a loss of smell/taste[43]). In contrast, unspecific symptom sets will require a high number of tests, especially in seasons where other respiratory conditions are prominent (currently, the fraction of SARS-CoV-2 cases among all influenza-like cases is less than 4%[44]). Random testing on a population level has the lowest positive rate in the regime of low prevalence that we focus on[41,45], but could be used in a targeted manner, e.g., screening of healthcare workers, highly vulnerable populations[10,46] or those living in the vicinity of localized outbreaks. We conclude that contact-tracing-based testing and highly specific symptoms-based testing should receive the highest priority, with the remaining test capacity used on less specific symptoms-based testing and random screening in particular settings.

The cooperation of the general population in maintaining a low reproduction number is essential even with efficient TTI strategies in place. Our results illustrate that the reproduction number in the hidden pool $R_t^H$—which reflects the public's behavior—is still central to disease control. Specifically, we found that $R_t^H \le 1.4$ (95 % CI, 1.23–1.69) can very likely be compensated by testing and isolating alone, whereas additional contract tracing shifts this boundary to $R_t^H \le 1.9$ (95% CI, 1.42–2.70, Supplementary Fig. 2 and Supplementary Table 1). Both of these values are substantially lower than the basic reproduction number of SARS-CoV-2, $R_0 \approx 3.3$[21,26,27]. Thus, if the goal is to contain the spread of SARS-CoV-2 with the available TTI-related resources, the reproduction number in the hidden pool will have to be reduced effectively by roughly 25–55% compared to the beginning of the pandemic. This effective reduction may be achieved by a suitable combination of hygiene measures, such as mask-wearing, filtering or exchanging contaminated air, and physical distancing. Useful accompanying measures voluntarily include: immediately and strictly self-isolating upon any symptoms compatible with COVID-19, avoiding travel to any region with a higher infection rate, keeping a personal contact diary, using the digital tracing app, selecting only those contacts that are essential for one's well being, and avoiding contacts inside closed rooms if possible. Most of these measures and also an efficient tracing cannot be achieved without the widespread cooperation of the population. This cooperation might be increased by a ramping up of coordinated educational efforts around explaining mechanisms and dynamics of disease spreading to a broad audience—instead of just providing behavioral advice.

The parameters of the model have been chosen to suit the situation in Germany. We expect our general conclusions to hold for other countries, but of course, parameters would have to be adapted to local circumstances. For instance, some Asia-Pacific countries can keep the spread under control, employing mainly test-trace-and-isolate measures[47]. Factors that contribute to this are (1) significantly larger investment in tracing capacity, (2) a smaller influx of externally acquired infections (especially in the case of new Zealand), and (3) the broader acceptance of mask-wearing and compliance with physical distancing measures. These countries illustrate that even once "control is lost" in the sense of our model, it can in principle be regained through political measures. A currently discussed mechanism to regain control is the "circuit breaker", a relatively strict lockdown to interrupt infection chains and bring case number down[48]. Such a circuit breaker or reset is particularly effective if it brings the system below the tipping point and thereby enables controlling the spread by TTI again. Therefore, it should be designed to keep a delicate balance between duration, stringency, and timeliness[49].

To conclude, based on a simulation of disease dynamics influenced by realistic TTI strategies with parameters taken from the example of Germany, we show that the spreading dynamics of SARS-CoV-2 can only be stabilized if effective TTI strategies are combined with hygiene and physical distancing measures that

**Table 2 Model variables.**

| Variable | Meaning | Units | Explanation |
|---|---|---|---|
| $H^a$ | Hidden asymptomatic pool | People | Non-traced, non-isolated people who are asymptomatic or avoid being tested |
| $H^s$ | Hidden symptomatic pool | People | Non-traced, non-isolated people who are symptomatic |
| $T^a$ | Traced asymptomatic pool | People | Known infected and isolated people who are asymptomatic |
| $T^s$ | Traced symptomatic pool | People | Known infected and isolated people who are symptomatic |
| $H$ | Hidden pool | People | Total non-traced people: $H = H^a + H^s$ |
| $T$ | Traced pool | People | Total traced people: $T = T^a + T^s$ |
| $N$ | New infections (traced and hidden) | Cases day$^{-1}$ | Given by: $N = \Gamma(\nu + \epsilon)R_t^H T + \Gamma R_t^H H + \Phi$ |
| $\hat{N}^{obs}$ | Observed new infections (influx to traced pool) | Cases day$^{-1}$ | Only cases of the traced pool; delayed on average by 4 days because of reporting |
| $\hat{R}_t^{eff}$ | Estimated effective reproduction number | - | Estimated from the cases of all pools: $\hat{R}_t^{eff} = N(t)/N(t-4)$ |
| $\hat{R}_t^{obs}$ | Observed reproduction number | - | The reproduction number that can be estimated only from the observed cases: $\hat{R}_t^{obs} = \hat{N}^{obs}(t)/\hat{N}^{obs}(t-4)$ |

keep the reproduction number in the general population below a value of approximately $R_t^H \leq 1.9$ (95% CI, 1.42–2.70). As a system stabilized by TTI with a finite capacity is only in a metastable state and can be tipped into instability by one-time effects, it would be desirable to keep a safety-distance even to these values, if possible. The above bounds on the reproduction number in the hidden pool can be easily recomputed for other countries with different TTI capacities and reproduction numbers.

## Methods

**Model overview**. We model the spreading dynamics of SARS-CoV-2 as the sum of contributions from two pools, i.e., traced $T$ and hidden $H$ infections (see the sketch in Fig. 1, and a complete list of parameters and variables, respectively in Tables 1 and 2). The first pool ($T$) contains traced cases revealed through testing or by contact tracing of an individual that has already been tested positive; all individuals in the traced pool are assumed to isolate themselves (quarantine), avoiding further contacts as well as possible. In contrast, in the second pool, infections spread silently and only become detected when individuals develop symptoms and get tested, or via random testing in the population. This second pool ($H$) is therefore called the hidden pool $H$; individuals in this pool are assumed to exhibit the behavior of the general population, thus of everyone who is not aware of being infected. We model the mean-field interactions between the hidden and the traced pool by transition rates, determining the timescales of the model dynamics. These transition rates can implicitly incorporate both the time course of the disease and the delays inherent to the TTI process, but we do not explicitly model delays between compartments. We distinguish between symptomatic and asymptomatic carriers—this is central when exploring different testing strategies (as detailed below). We also include effects of non-compliance and imperfect contact tracing, as well as a nonzero influx $\Phi$ of new cases that acquired the virus from outside. As this influx makes the eradication of SARS-CoV-2 impossible, only an exponential growth of cases or a stable rate of new infections is possible modeling outcomes. Given the two possible behaviors of the system, indefinite growth, or stable cases, we frame our investigation as a stability problem. The aim is to implement test-trace-and-isolate strategies to allow the system to remain stable.

**Spreading dynamics**. Concretely, we use a modified SIR-type model, where infections $I$ are either symptomatic ($I^s$) or asymptomatic ($I^a$), and they belong to the hidden ($H$) or a traced ($T$) pool of infections (Fig. 1), thus creating in total four compartments of infections ($H^s$, $H^a$, $T^s$, $T^a$). New infections are asymptomatic with a ratio $\xi^{ap}$; the others are symptomatic. In all compartments, individuals are removed with a rate $\Gamma$ because of recovery or death (see Table 1 for all parameters).

In the hidden pool, the disease spreads according to the reproduction number $R_t^H$. This reproduction number reflects the disease spread in the general population, without testing induced isolation of individuals. In addition, the hidden pool receives a mobility-induced influx $\Phi$ of new infections. Cases are removed from the hidden pool (i) when detected by TTI, and put into the traced pool, or (ii) due to recovery or death.

The traced pool $T$ contains those infected individuals who have been tested positive as well as their positively tested contacts. As these individuals are (imperfectly) isolated, they cause infections with a rate $\nu \Gamma R_t^H$, which are subsequently isolated and therefore stay in the traced pools and additional infections with a rate $\epsilon \Gamma R_t^H$, which are missed and act as an influx to the hidden pools. $\nu$ is the isolation factor, and $\epsilon$ is the leak factor. The overall reproduction number of the traced pool is therefore $R_t^T = (\nu + \epsilon)R_t^H$.

In the scope of our model, it is important to differentiate exchanges from pool to pool that are based either on the "reassignment" of individuals or on infections. To the former category belongs the testing and tracing, which transfer cases from the hidden pool to the traced pool. These transfers involve a subtraction and addition of case numbers in the respective pools. To the latter category belongs the recurrent infections $\Gamma R_t^H$ or $\nu \Gamma R_t^H$ and the 'leak' infections $\epsilon \Gamma R_t^H$. Exchanges of this category involve only the addition of case numbers in the respective pool.

Within our model, we concentrate on the case of low incidence and a low fraction of immune people, as in the early phase of any new outbreak. Our model can also reflect innate or acquired immunity; one must rescale the population or the reproduction number. The qualitative behavior of the dynamics is not expected to change.

**Parameter choices and scenarios**. For any testing strategy, the fraction of infections that do not develop any symptoms across the whole infection timeline is an important parameter, and this also holds for testing strategies applied to the case of SARS-CoV-2. In our model, this parameter is called $\xi^{ap}$ and includes, besides the real asymptomatic infections $\xi$, the fraction of individuals that avoid testing $\varphi$.

The exact value of the fraction of asymptomatic infections $\xi$, however, is still fraught with uncertainty, and it also depends on age[15,50,51]. While early estimates were as high as 50 % (for example ranging from 26 to 63%[52]), these early estimates suffered from reporting bias, small sample sizes and sometimes included pre-symptomatic cases as well[22,53]. Recent bias-corrected estimates from large sample sizes range between 12%[22] and 33%[23]. We decided to use 15% for the pure asymptomatic ratio $\xi$.

In addition, we include a fraction $\varphi$ of individuals avoiding testing. This can occur because individuals do not want to be in contact with governmental authorities or because they deem risking a spread of SARS-CoV-2 less important than having to quarantine[16]. As this part of the population may act in the same manner as asymptomatic persons, we include it in the asymptomatic compartment of the hidden pool, assuming a value of 0.2. We thus arrive at an effective ratio of asymptomatic infections $\xi^{ap} = \xi + (1 - \xi)\varphi = 0.32$. We assume that both symptomatic and asymptomatic persons have the same reproduction number.

In general, infected individuals move from the hidden to the traced pool after being tested; yet, a small number of infections will leak from the traced to the hidden pool with rate $\epsilon \Gamma R_t^H$, with $\epsilon = 0.1$. A source of the leak would be a contact that has been infected, traced, and tested positive but still ignores quarantine instructions. For the model, this individual has the same effect on disease dynamics as someone from the hidden pool.

Another crucial parameter for any TTI strategy is the reproduction number in the hidden pool $R_t^H$. This parameter that typically represents the main driver of the spreading dynamics is, by definition, impossible to measure. It depends mainly on the contact behavior of the population and ranges from $R_0$ in the absence of contact restrictions to values below 1 during strict lockdown[2]. For the default parameters of our model, we used a value of $R_t^H = 1.8$. This parameter was chosen after all others, aiming to mirror the epidemic situation in Germany during the early summer months, when infections remained approximately constant. It is just below the critical value $R_{crit}^H = 1.98$ for the default scenario, hence $\hat{R}_t^{eff} = 1$. This value of $R_t^H = 1.8$ is ~54% lower than the basic reproduction number $R_0 \approx 3.3$. Hence, we assume that some non-pharmacological interventions (physical distancing or hygiene measures) are in place, as was the case in Germany during the early summer months[1,2]. For additional scenarios, we explored the impact of both higher and lower values of $R_t^H$ on our TTI strategy (see Figs. 7, 8 and Supplementary Fig. 2).

**Testing-and-tracing strategies**. We consider three different testing-and-tracing strategies: random testing, symptom-driven testing, and specific testing of traced

contacts. Despite the naming—chosen to be consistent with existing literature[4,36,42,54,55]— isolation of the cases tested positive is part of all of these strategies. The main differences lie in whom the tests are applied to and whether past contacts of an infected person are traced and told to isolate. Our model simulates the parallel application of all three strategies—as it is typical for real-world settings, and yields the effects of the "pure' application of these strategies as corner cases realized via specific parameter settings.

Random testing is defined here as applying tests to individuals irrespective of their symptom status or whether they belonged to the contact-chain of other infected individuals. In our model, random testing transfers infected individuals from the hidden to the traced pool with a fixed rate $\lambda_r$, irrespective of them showing symptoms or not. In reality, random testing is often implemented as situation-based testing for a sub-group of the population, e.g., at a hot-spot, for groups at risk, or for people returning from travel. Such situation-based strategies would be more efficient than the random testing assumed in this model. Nonetheless, because random testing can detect symptomatic and asymptomatic persons alike, we decided to evaluate its potential contribution to containing the spread.

The number of random tests that can be performed is limited by the available laboratory and sample collection capacity. For orientation, we included therefore a maximal testing capacity of $\lambda_{r,\max} = 0.002$ test per person and day, which reflects the laboratory capacity in Germany (1.2 Mio. per week)[56,57]. Potentially, the testing capacity can be increased by pooling PCR tests, without strongly reducing the sensitivity[25]. We acknowledge this possibility by taking into account a ten times larger testing capacity, $10 \cdot \lambda_{r,\max} = 0.02$. This would correspond to every person being tested on average every 50 days (7 weeks)—summing to about 12 Mio. tests per week in Germany.

Symptom-driven testing is defined as applying tests to individuals presenting symptoms of COVID-19. In this context, it is important to note that non-infected individuals can have symptoms similar to those of COVID-19, as many symptoms are rather unspecific. Although symptom-driven testing suffers less from imperfect specificity, it can only uncover symptomatic cases that are willing to be tested (see below). Here, "symptomatic infected individuals' are transferred from the hidden to the traced pool at rate $\lambda_s$.

We define $\lambda_s$ as the daily rate at which symptomatic individuals get tested, among the subset who are willing to get tested. As the default value, we use $\lambda_s = 0.1$, which means that one in ten people that show symptoms gets tested each day and are subsequently isolated. Testing and isolation happen immediately in this model, but their report into the observed new daily cases $\hat{N}^{\mathrm{obs}}$ is delayed. Further real-world delays can effectively be modeled by a lower effective $\lambda_s$. In theory, this rate could be increased to one per day. However, this parameter range is on purpose, not simulated here. For SARS-CoV-2, such a fast detection is unrealistic because typically infected people show a delay of 1–2 days between the beginning of infectiousness and showing symptoms[58]. Hence, $\lambda_s \approx 0.5$ is an upper limit to the symptom-driven testing rate.

Tracing contacts of positively tested individuals presents a very specific test strategy and is expected to be effective in breaking the infection chains if contacts self-isolate sufficiently quickly[4,42,59]. However, as every implementation of a TTI strategy is bound to be imperfect, we assume that only a fraction $\eta < 1$ of all contacts can be traced. These contacts, if tested positive, are then transferred from the hidden to the traced pool. No delay is assumed here. The parameter $\eta$ effectively represents the fraction of secondary and tertiary infections found through contact tracing. As this fraction decreases when the delay between testing and contact tracing increases, we assumed a default value of $\eta = 0.66$, i.e., on average, only two-thirds of subsequent infections are prevented.

Contact tracing is mainly done by the health authorities in Germany, and this clearly limits the maximum number $N_{\max}$ of observed new cases $\hat{N}^{\mathrm{obs}}$, for which contact tracing is still functional. In the first part of the manuscript, we assume for simplicity that $\hat{N}^{\mathrm{obs}}$ is sufficiently small to not exceed the tracing capacity; in the second part, we explicitly explore the role of this limit.

In principle, the tracing capacity limit can be expressed in two ways, either as the number of observed cases $\hat{N}^{\mathrm{obs}}$, at which tracing starts to break down (denoted by $N_{\max}$), or as number of positive contacts that can maximally be detected and handled on average by the health departments ($n_{\max}$). Both values depend strongly on the personnel capacity of the health departments and the population's contact behavior. From the system's equilibrium equations, we derive a linear relation between the two, with the proportionality being a function of the epidemiological and TTI parameters (Supplementary Eq. (14)). For simplicity, we only use $N_{\max}$ in the main text and refer the interested reader to the derivation in Supplementary Note 2.

As a default value, we assume $n_{\max} = 300$ positive contacts that can be handled per day. This corresponds to $N_{\max} = 718$ observed cases per day, from which the above-mentioned 300 cases were found through contact tracing. Thus, the remaining 418 either originate within the traced pool (e.g., infected family members) or were found through symptom-based testing and are therefore considered to be detected with much less effort. This limit of $n_{\max} = 300$ is currently well within reach of the 400 health departments in Germany. At first sight, this limit may appear low (about one case per working day per health department). However, identifying, contacting, and counseling all contact persons

(thus many more persons than 300), and finally testing them and controlling their quarantine requires considerable effort.

Any testing can, in principle, produce both false-positive (quarantined individuals who were not infected) and false-negative (non-quarantined infected individuals) cases. In theory, false-positive rates should be meager (0.2% or less for RT-PCR tests). However, testing and handling of the probes can induce false-positive results[60,61]. Under the low prevalence of SARS-CoV-2, false-positive could therefore outweigh true-positive, especially for the random-testing strategy, where the number of tests required to detect new infections would be very high[62,63]. This should be carefully considered when choosing an appropriate testing strategy but has not been explicitly modeled here, as it does not contribute strongly to whether or not the outbreak could be controlled.

**Model equations**. The contributions of the spreading dynamics and the TTI strategies are summarized in the equations below. They govern the spreading dynamics of case numbers in and between the hidden and the traced pool, $H$ and $T$. We assume a regime of low prevalence and low immunity, i.e., the majority of the population is susceptible. Thus, the dynamics are completely determined by spread (represented by the reproduction numbers $R_t$), recovery (characterized by the recovery rate $\Gamma$), external influx $\Phi$ and the impact of the TTI strategies:

$$\frac{dT}{dt} = \underbrace{\Gamma(\nu R_t^H - 1)T}_{\text{spreading dynamics}} + \underbrace{\lambda_s H^s + \lambda_r H}_{\text{testing}} + \underbrace{f(H^s, H)}_{\text{tracing}}, \quad (1)$$

$$\frac{dH}{dt} = \underbrace{\Gamma(R_t^H - 1)H}_{\text{spreading dynamics}} - \underbrace{(\lambda_s H^s + \lambda_r H)}_{\text{testing}} - \underbrace{f(H^s, H)}_{\text{tracing}} + \underbrace{\Gamma \epsilon R_t^H T}_{\text{missed contacts}} + \underbrace{\Phi}_{\text{external influx}}, \quad (2)$$

$$\frac{1}{1-\xi^{\mathrm{ap}}}\frac{dH^s}{dt} = \underbrace{\Gamma\left(R_t^H H - \frac{H^s}{1-\xi^{\mathrm{ap}}}\right)}_{\text{spreading dynamics}} - \underbrace{\frac{(\lambda_s + \lambda_r)H^s}{1-\xi^{\mathrm{ap}}}}_{\text{testing}}$$
$$- \underbrace{f(H^s, H)}_{\text{tracing}} + \underbrace{\Gamma \epsilon R_t^H T}_{\text{missed contacts}} + \underbrace{\Phi}_{\text{external influx}}, \quad (3)$$

$$H^a = H - H^s, \quad (4)$$

with

$$f(H^s, H) = \min\left\{n_{\max}, \eta R_t^H(\lambda_s H^s + \lambda_r H)\right\}. \quad (5)$$

Equations (1) and (2) describe the dynamical evolution of both the traced and hidden pools. However, they are not sufficient to completely describe the underlying dynamics of the system in the hidden pool, as the symptomatic and asymptomatic sub-pools behave slightly differently: only from the symptomatic hidden pool ($H^s$) cases can be removed because of symptom-driven testing. Thus the specific dynamics of $H^s$ is defined by equation (3). The dynamics of the asymptomatic hidden pool ($H^a$) can be inferred from Eq. (4). In the traced compartment, the asymptomatic and symptomatic pools do not need to be distinguished, as their behavior is assumed to be identical. Equation (5) reflects a potential limit $n_{\max}$ of the tracing capacity of the health authorities. It is expressed as the total number of positive cases that can be detected from tracing the contacts of people detected via symptom-driven testing (from $H^s$) or via random testing (from $H$).

**Central epidemiological parameters that can be observed**. In the real world, the disease spread can only be observed by the traced pool. While the "true" number of daily infections $N$ is a sum of all new infections in the hidden and traced pools, the "observed" number of daily infections $\hat{N}^{\mathrm{obs}}$ is the number of new infections in the traced pool delayed by a variable reporting delay $\alpha$. This includes internal contributions and contributions from testing and tracing:

$$N(t) = \underbrace{\Gamma(\nu + \epsilon)R_t^H T(t)}_{\text{traced pool}} + \underbrace{\Gamma R_t^H H(t)}_{\text{hidden pool}} + \underbrace{\Phi}_{\text{external influx}} \quad (6)$$

$$\hat{N}^{\mathrm{obs}}(t) = \left[\underbrace{\Gamma\nu R_t^H T(t)}_{\text{traced pool}} + \underbrace{\lambda_s H^s(t) + \lambda_r H(t)}_{\text{testing}} + \underbrace{f(H^s(t), H(t))}_{\text{tracing}}\right] \mathcal{G}[\alpha = 4, \beta = 1](t), \quad (7)$$

where $f(H^s, H)$ is defined in (5), ⊛ denotes a convolution and $\mathcal{G}$ a Gamma distribution that models a variable reporting delay. The spreading dynamics are usually characterized by the observed reproduction number $\hat{R}_t^{\mathrm{obs}}$, which is calculated from the observed number of new cases $\hat{N}^{\mathrm{obs}}(t)$. We here use the definition underlying the estimates that are published by Robert-Koch-Institute, the official body responsible for epidemiological control in Germany[64]: the reproduction number is the relative change of daily new cases $N$ separated by 4 days (the

assumed serial interval of COVID-19[65]):

$$\hat{R}_t^{\text{obs}} = \frac{\hat{N}^{\text{obs}}(t)}{\hat{N}^{\text{obs}}(t-4)} \tag{8}$$

$$\hat{R}_t^{\text{eff}} = \frac{N(t)}{N(t-4)} \tag{9}$$

While only $\hat{R}_t^{\text{obs}}$ is accessible from the observed new cases, in the model, one can also define an effective reproduction number $\hat{R}_t^{\text{eff}}$ from the total number of daily new infections.

In contrast to the original definition of $\hat{R}_t^{\text{obs}}$ [64], we do not need to remove real-world noise effects by smoothing this ratio.

**Numerical calculation of solutions and critical values.** The numerical solution of the differential equations governing our model was obtained using a versatile solver based on an explicit Runge–Kutta (4,5) formula, `@ode45`, implemented in MATLAB (version 2020a), with default settings. This algorithm allows the solution of non-stiff systems of differential equations in the shape $y' = f(t, y)$, given a user-defined time-step (for us, 0.1 days). Suitability and details on the algorithm are further discussed in ref. [66].

To derive the tipping point between controlled and uncontrolled outbreaks (e.g., critical values of $R_t^H$), and to plot the stability diagrams, we used the `@fzero` MATLAB function. This function uses a combination of bisection, secant, and inverse quadratic interpolation methods to find the roots of a function. For instance, following the discussion of Supplementary Note 1, $R_{\text{crit}}$ was determined by finding the roots of the function returning the real part of the linear system's largest eigenvalue.

**Reporting summary**. Further information on research design is available in the Nature Research Reporting Summary linked to this article.

## Data availability
Data used in this study was obtained through numerical simulation. It is available together with the code for solving our model's equations for default and user-customized parameters at https://github.com/Priesemann-Group/covid19_tti (https://doi.org/10.5281/zenodo.4290679). Alternatively, an interactive platform for simulating scenarios different from the herein presented is available on http://covid19-tti.ds.mpg.de, and users may download the data generated.

## Code availability
We provide the code for generating graphics and all the different analyses included in both this manuscript and its Supplementary Information at https://github.com/Priesemann-Group/covid19_tti (https://doi.org/10.5281/zenodo.4290679). An interactive platform for simulating scenarios different from the herein presented is available on http://covid19-tti.ds.mpg.de.

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

## Acknowledgements

We thank the Priesemann group for exciting discussions and their valuable comments. We also thank helpful comments and suggestions from Jakob Ruess (Inria), Ralf Meyer (Göttingen Uni), Álvaro Olivera-Nappa (Universidad de Chile). Open Access funding enabled and organized by Projekt DEAL. All authors received support from the Max-Planck-Society. S.C. acknowledges funding from the Centre for Biotechnology and Bioengineering - CeBiB (PIA project FB0001, Conicyt, Chile). M.L., J.D., and P.S. acknowledge funding by SMARTSTART, the joint training program in computational neuroscience by the VolkswagenStiftung and the Bernstein Network. JZ received financial support from the Joachim Herz Stiftung. M. Wibral is employed at the Campus Institute for Dynamics of Biological Networks funded by the VolkswagenStiftung.

## Author contributions

S.C., J.D., J.Z., and V.P. designed research. S.C. conducted research. S.C., J.D., J.Z., M.L., M. Wibral, M. Wilczek, and V.P. analyzed the data. S.C., P.S., M.L., J.U., and S.B.M. created figures. All authors wrote the paper.

## Funding

## Competing interests

The authors declare no competing interests.
