## [Peer Review File · Nature Communications]

REVIEWER COMMENTS

Reviewer #1 (Remarks to the Author):

Review for " The challenges of containing SARS-CoV-2 via test-trace-and-isolate"

Overview. Using a compartmental model, the authors model SARS-CoV2 cases that are successfully tested, traced and isolated, as well as a hidden pool of cases that have not been tested and traced. Using this model, the authors explore the dynamics of the outbreak when different testing strategies are used (symptom-based vs random), and the effect of the TTI capabilities in controlling the spread. The authors conclude that even with efficient TTI strategies in place, the reproduction number in the hidden pool must be maintained well below the R_0 for SARS-CoV2 to control its spread.

Recommendation. The conditions that must be achieved before the current restrictions in place to control the spread of COVID-19 can be relaxed is one of the most important public health questions at the moment. I congratulate the authors on tackling this extremely important scientific challenge. I found the manuscript to be well-written and well-argued and thoroughly enjoyed reading it. The model incorporates sufficient real-world complexity to serve a useful purpose. I recommend that the manuscript be published with minor revisions.

Detailed Recommendations.

Major Recommendations

1. Line 134. An influx of 4000 cases is quite large relative to the limits of tracing capacity chosen by the authors. Was there a reason for selection this very large number? It would be interesting to see the effects if the influx of cases is 0.5, 1 and 1.5 times the maximum tracing capacity. I would also like to know over how many days was this influx concentrated (it is not clear to me from the graph).

As a potentially additional analyses, the authors could illustrate periodic influx of a certain number of cases, say every 60 days, although this last point is less important.

2. I do think that the manuscript is a bit long and some material can be moved to Supplementary Information without comprising its readability. I would perhaps move Figure 4 to a supplementary text.

Minor Recommendations

Abstract Line 10. Perhaps the author meant to use "cooperation" rather than cooperativity as the latter has a technical meaning? If " cooperativity " was intended, I suggest authors rephrase as the word is not commonly used in epidemiological literature as far as I am aware.

Line 32. Rephrase, as " surfacing widely distributed" is grammatically awkward.

Line 88 Missing word after section.

Line 103. Missing word after solid.

Line 109. It would be helpful if the authors could define "meta-stable" before it is first used in the caption of Figure 4.

Line 157, "This ramping is due to exceeding the tracing capacity N_{max} , and the spread starts to accelerate". Consider rephrasing as this sentence reads a bit awkward with the change in tense in the first and second parts of the sentence.

Figure 7 caption. "grey lines in plots" Start with a capital letter.

Line 476. "This corresponds to uncovering 300 positive contact persons...." Can the authors elaborate this breakdown please?

Reviewer #2 (Remarks to the Author):

In this manuscript, Contreras et al. evaluate the factors that influence whether 'test-trace-and-isolate' (TTI) is sufficient to contain SARS-CoV-2 transmission, even when there is a steady influx of new cases into the population of interest. Control is modeled as isolation of positive cases identified via random testing, syndromic testing and/or contact tracing. They find two tipping points for uncontrolled spread. One is when the reproduction number for the general population remains too high despite preventive measures such as social distancing. The second is when the capacity of the testing and tracing program is exceeded.

Overall, I am very impressed with this manuscript. The writing is very clear and thorough, and the figures convey a lot of information. The motivation for the analyses is laid out well, the methodology strikes a nice balance between simplicity and inclusion of complex transmission dynamics, explicit equations and access to code ensures reproducibility, the results are thoroughly described, and the discussion describes several important implications for public health. Meaningful findings include resource limitations lead to metastable dynamics, TTI can work when preventative measures are in place but not for R of 3.3, and the observed R may underestimate the true R . However, I would recommend two improvements prior to acceptance into a high-profile journal. The first is to shorten the manuscript as there is a fair amount of redundancy and thus some of the key material may be hard for readers to find. The second is to provide some additional clarifications and reflections on the methods (detailed below).

Feedback on methodology:

- Lines 394/417: There seem to be slightly different definitions of epsilon here. In particular I think there is a distinction between a symptomatic case that does not go into the T pool and a positive contact that doesn't go into the T pool (since the latter can be asymptomatic)?
- Line 395: Strictly speaking it seems that $R = [(1 - e)v + e]R_{tH}$ rather than $(v + e)R_{tH}$?
- Line 458: Can you mention whether you are assuming that testing yields instantaneous results? Or are results delayed, but folks are immediately isolated (seems more relevant for contact tracing)?
- Line 467: Is there a delay between the time that a case is isolated and when the cases' contacts are isolated? Or is it assumed to be instantaneous?
- Equation 3: It looks like there is an assumption that the proportion of traced H that are H_s and the missed contacts that are H_s are both equal to $(1 - \text{asymptomatic-ratio})$. But it strikes me that it might be less than this because symptomatic cases are more likely than asymptomatic cases to be in the T pool?
- Equation 5: It seems that contacts of positive contacts should also be traced (e.g. multiple generations of contact tracing)? Is this reflected in $f(\cdot)$? On the flip side, it seems that some of the contacts might already be in the T pool and that could decrease the value of $f(\cdot)$? Can you comment / clarify?
- Equation 9: Regarding R_{eff} : Since the influx does not represent transmission, I wonder if it should be subtracted from the numerator?
- Table 2: Do H_a and H_s necessarily reflect non-traced individuals? Or just non-isolated (e.g. can there be non-isolated, traced individuals that are part of H)? Similarly, do T_a and T_s necessarily reflect traced individuals, or just isolated ones?

Minor editing suggestions:

- Line 32 - 'surfacing' - awkward wording
- Lines 321-324: I found these sentences unclear.
- Line 397 Consider eliminating 'quickly waning immunity' as it is not relevant to the model
- Line 424: Unclear what is meant by 'This value causes new infections to be approximately constant' with $R = 1.8$
- Line 457: Would it be more accurate to say λ_s is rate in which symptomatic individuals

get tested, amongst the subset who are willing to get tested?

- Line 530: Typo: which section?
- Fig 1A/B – Beautiful aesthetics, but hard to understand the details without a figure legend. Also, unclear how this adds value to figure 2
- Fig 2B – Lacking legend
- Fig 3 Legend – slightly confusing wording: Does (B,E) correspond to NEW infections only?
- Fig 5 Hard to see the difference between C and F. Would rescaling help?
- Figure S2: I like this figure. However, it isn't clear whether decreasing/increasing a parameter ends up decreasing or increasing RH_{crit} . I wonder if A) would be more informative if each panel was simply a line graph of how the RH_{crit} changed for particular values of the parameter being considered (rather than show how a distribution of parameters is mapped).
- Table 1: Is $R_0 = 3.3$ used in the model? If not, I'd remove it from the table. Or perhaps it should be listed as a special case of RH_t ? (And my apologies if I missed where R_0 was included in the model)

Personal observation (no changes needed):

- Lines 159-161. Interesting point. I wonder if you can deduce R_{hidden} , from R_{Obs} and the fraction of cases identified from symptomatic screening vs contact tracing?

Reviewer #3 (Remarks to the Author):

This was interesting and well-written modelling study on test-trace-isolate strategies to control SARS-COV-2. The model makes a number of simplifying assumptions, especially about contact patterns, but has a nice structure in considering "hidden" and "traced" pools. The model and methods are very clearly laid out and well explained, and the authors should be commended on their sharing of code at the review stage.

The main finding that TTI is going to be most effective when combined with other physical distancing measures is perfectly sensible, and this is supported by other modelling and empirical studies. However, my main concern is that the rather negative message about TTI approaches in places extends beyond the scope of the current model, and is dependent on some key assumptions. To some extent this may be due to the focus in parameters relevant to Germany, but as the messages are intended to be broader it is important that these limitations are very clear.

Firstly, as far as I can see all of the scenarios assume a constant (and reasonably high) influx of new cases. This is a specific scenario, and one in which TTI is likely to be less effective. When rates of influx into the population are low (e.g. with efficient border control/testing) I think that the sensitivity of the main results to this parameter should be explored in more detail.

Secondly, as the authors briefly acknowledge in their discussion, random testing is likely to be most efficient when implemented in specific settings. It is not especially surprising that it doesn't work very well when applied on a population scale. Given this, I think the messaging needs to be very clear about the limited scope of this model for testing the efficacy of random testing in the ways it is likely to be implemented in the real world.

Specific comments:

Lines 9-11, this is not how I read the results - suggest expanding a little to say that likely success of TTI is dependent on the reproduction number

Lines 32-33: Not sure this is true (lots of evidence of clustered transmission) - can you elaborate/reference?

Line 39: This might be true for some places, but not others (e.g. NZ) - suggest toning this

down/rephrasing

Lines 134-135: this seems like a very large instantaneous influx - would a smaller number not be more realistic/sensible to use here?

Line 171 (and figure 7): While it can be useful to tease apart the contribution of random/symptomatic testing and contact tracing, in reality all three will be used in conjunction. I think perhaps, therefore, that the conclusions about random testing (line 188-191) are a too pessimistic.

Line 266: base reproduction number needs a citation

Lines 330-340: Suggest a bit of a rewrite here. Random testing is, as you say, likely to be important in specific settings such as hospitals/schools/universities. You should acknowledge that your study can't capture this, and be careful not to come across as though these settings are not important.

Lines 405-409: 12% is very much on the low end of rates of asymptomatic individuals. How does increasing this assumption affect your results?

Lines 414-415: this assumption may not be realistic. Do your results hold if you assume that asymptomatic patients are e.g. half as infectious as symptomatics?

Reviewer 1

Overview. Using a compartmental model, the authors model SARS-CoV2 cases that are successfully tested, traced and isolated, as well as a hidden pool of cases that have not been tested and traced. Using this model, the authors explore the dynamics of the outbreak when different testing strategies are used (symptom-based vs random), and the effect of the TTI capabilities in controlling the spread. The authors conclude that even with efficient TTI strategies in place, the reproduction number in the hidden pool must be maintained well below the R_0 for SARS-CoV2 to control its spread.

Recommendation. The conditions that must be achieved before the current restrictions in place to control the spread of COVID-19 can be relaxed is one of the most important public health questions at the moment. I congratulate the authors on tackling this extremely important scientific challenge. I found the manuscript to be well-written and well-argued and thoroughly enjoyed reading it. The model incorporates sufficient real-world complexity to serve a useful purpose. I recommend that the manuscript be published with minor revisions.

We thank the reviewer for their concise summary of our manuscript and are pleased with their positive evaluation and their joy when reading.

Major Recommendations

1. Line 134. An influx of 4000 cases is quite large relative to the limits of tracing capacity chosen by the authors. Was there a reason for selection this very large number? It would be interesting to see the effects if the influx of cases is 0.5, 1 and 1.5 times the maximum tracing capacity. I would also like to know over how many days was this influx concentrated (it is not clear to me from the graph). As a potentially additional analyses, the authors could illustrate periodic influx of a certain number of cases, say every 60 days, although this last point is less important.

Even though this number seems to be quite large, we would like to remark that it is the **total number** of externally acquired infections entering over several days, not the **observed number per day**. We were inspired by the large number of people in Germany that came back from summer holiday, which led to a transient increase in cases similar to that in our "Default" scenario (July-August). From this perspective, the influx is rather low: during the first two weeks of August, 900 new cases were detected at test centres covering 3 out of 6 highway border crossings located in the state of Bavaria alone [1]. Given that cars account for only 20% of long-distance holidays [2], one could estimate 4500 observed cases in a single week in August for those crossing Bavarian borders alone - only a lower bound for the number of hidden infections among holidaymakers returning to all of Germany. We now explain this inspiration in the text and also state the number of days that this influx was concentrated over (lines 117–122):

As an example of how this limited tracing capacity can cause a new tipping point to instability, we simulate here a short but large influx of externally acquired infections (a total of 4000 hidden cases with 92% occurring in the 7 days around $t = 0$, normally distributed with $\sigma = 2$ days, see Fig. 3). This exemplary influx is inspired from the large number of German holidaymakers returning from summer vacation, and is a rather conservative estimate given that there were 900 such cases observed in the first two weeks of July at Bavarian highway test-centres alone [1].

Furthermore, we show three additional influx scenarios: we varied the amplitude of the influx (Supplementary Figure 3), its duration (Supplementary Figure 4), and included periodicity (new Figure 4). Note that the influx increases the number of hidden cases, while the maximal tracing capacity N_{\max} represents the **maximum observed number of cases** \hat{N}^{obs} - therefore, a "peak influx" of N_{\max} does not overwhelm the tracing capacity, as seen in Supplementary Figure 3c,d. We thank the reviewer for suggesting periodic input, and now highlight its potential impact in the main manuscript: if the period is chosen small enough, their cumulative impact causes a buildup of cases that can overwhelm the tracing capacity. The relevant sentences in the main text read (lines 128–134):

Not only the capacity limit, but also the amplitude of the influx (Supplementary Fig. 3), its duration (Supplementary Fig. 4) or whether it occurs periodically (Fig. 4) can decide whether the observed new cases \hat{N}^{obs} exceed the capacity limit N_{\max} and cause a tipping-over into instability. In particular, periodic influxes (as holidays) may cause the tipping-over not necessarily because of a single event but due to their cumulative impact. These scenarios demonstrate that the limited tracing capacity renders the system meta-stable: if the capacity limit is exceeded due to some external perturbation, the tracing cannot compensate the perturbation and the spread gets out of control.

2. I do think that the manuscript is a bit long and some material can be moved to Supplementary Information without comprising its readability. I would perhaps move Figure 4 to a supplementary text.

We agree that our manuscript became quite lengthy. To reduce redundancy, we simplified Figure 4 (resulting in new Fig. 5), and moved the original Figure 4 as well as the accompanying text to the Supplementary Information. We also shortened the part of the discussion that deals with the different testing strategies.

Minor Recommendations Abstract Line 10. Perhaps the author meant to use "cooperation" rather than cooperativity as the latter has a technical meaning? If "cooperativity" was intended, I suggest authors rephrase as the word is not commonly used in epidemiological literature as far as I am aware.

We adopted "cooperation", and also rephrased the whole abstract. The sentence now reads (lines 8–10):

We investigated how these tipping points depend on challenges like limited cooperation, missing contacts, and imperfect isolation.

Line 32. Rephrase, as "surfacing widely distributed" is grammatically awkward.

We rephrased the sentence, it now reads (lines 33–35):

Furthermore, SARS-CoV-2 infections generally appear throughout the whole population (not only in regional clusters), which hinders an efficient and quick implementation of TTI strategies.

Line 88 Missing word after section.

Line 103. Missing word after solid.

We thank the reviewer for spotting these, and filled in the missing words.

Line 109. It would be helpful if the authors could define "meta-stable" before it is first used in the caption of Figure 4.

It is now defined in the main text (lines 132–134) :

These scenarios demonstrate that the limited tracing capacity renders the system meta-stable: if the capacity limit is exceeded due to some external perturbation, the tracing cannot compensate the perturbation and the spread gets out of control.

Line 157, "This ramping is due to exceeding the tracing capacity N_{max} , and the spread starts to accelerate". Consider rephrasing as this sentence reads a bit awkward with the change in tense in the first and second parts of the sentence.

We rephrased this sentence, it now reads (lines 150–151):

This ramping is due to the tracing capacity N_{max} being exceeded, which causes an acceleration of the spread.

Figure 7 caption. "grey lines in plots" Start with a capital letter.

Corrected as suggested.

Line 476. "This corresponds to uncovering 300 positive contact persons..." Can the authors elaborate this breakdown please?

We revised the exact equation that relates the number of contacts that can be traced n_{\max} and the number of observed cases at which this tracing capacity is reached N_{\max} , and improved upon the explanation in the methods section.

The revision was motivated by the fact that we found a small discrepancy of about 5% between the number of cases where the system lost stability in the simulations and the analytical equation that we used to calculate N_{\max} when scanning the peak amplitude of the external influx (Supplementary Figure 3). Therefore, the N_{\max} values throughout the whole manuscript have been updated (see Figures 3 and 4 as well as Supplementary Figures 3 and 4).

The new text now reads (lines 466–481 and 738–750):

Tracing.[...] In principle, the tracing capacity limit can be expressed in two ways, either as the number of observed cases \hat{N}^{obs} , at which tracing starts to break down (denoted by N_{\max}), or as number of positive contacts that can maximally be detected and handled on average by the health departments (n_{\max}). Both values depend strongly on the personnel capacity of the health departments and the population's contact behavior. From the system's equilibrium equations, we derive a linear relation between the two, with the proportionality being a function of the epidemiological and TTI parameters (Supplementary Equation 14). For simplicity, we only use N_{\max} in the main text and refer the interested reader to the derivation in Supplementary Information Section 3.

As a default value, we assume $n_{\max} = 300$ positive contacts that can be handled per day. This corresponds to $N_{\max} = 718$ observed cases per day, from which the above-mentioned 300 cases were found through contact tracing and the remaining 418 either originate within the traced pool (e.g. infected family members), or were found through symptom-based testing and are therefore considered to be detected with much less effort. This limit of $n_{\max} = 300$ is currently well within reach of the 400 health departments in Germany. At first sight, this limit may appear low (about one case per working day per health department). However, identifying, contacting and counselling all contact persons (thus many more persons than 300), and finally testing them and controlling their quarantine requires considerable effort.

Equilibrium equations for case numbers above tracing capacity. We can also derive equilibrium equations for the case where tracing capacity is exceeded ($\eta\lambda_s R_t^H H_\infty^s > n_{\max}$). Remark that in this case, that critical reproduction number at which the equilibrium is stable, is smaller than for the $\eta\lambda_s R_t^H H_\infty^s < n_{\max}$ case.

When the tracing capacity is exceeded, the values returned by function $f(H^s, H)$ (defined by equation 5 in the main manuscript) are constant $f(H^s, H) = n_{\max}$. Then, setting the equations 1-3 of the main manuscript equal to zero leads to:

$$\begin{aligned} T_\infty &= \frac{\lambda_s H_\infty^s + n_{\max}}{\Gamma (1 - \nu R_t^H)} \\ H_\infty &= H_\infty^s \frac{\lambda_s}{\Gamma} \left(\frac{\frac{\Gamma}{\lambda_s} + \xi^{\text{ap}}}{1 - \xi^{\text{ap}}} \right) \\ \lambda_s H_\infty^s &= \frac{n_{\max} \left(\frac{\epsilon R_t^H}{\nu R_t^H - 1} + 1 \right) - \Phi}{(R_t^H - 1) \left(\frac{\frac{\Gamma}{\lambda_s} + \xi^{\text{ap}}}{1 - \xi^{\text{ap}}} \right) - \left(1 - \frac{\epsilon R_t^H}{1 - \nu R_t^H} \right)} \end{aligned}$$

Similarly, we can derive an equation for N_{\max} , which represents the **maximum observed number of cases** at the tracing capacity limit, by using the new equilibrium values and the tracing-limit condition $f(H^s, H) = n_{\max}$ in equation 2:

$$N_\infty^{\text{obs}} = \frac{(R_t^H - 1) \left(\frac{\frac{\Gamma}{\lambda_s} + \xi^{\text{ap}}}{1 - \xi^{\text{ap}}} \right) n_{\max} - \Phi}{(R_t^H - 1) \left(\frac{\frac{\Gamma}{\lambda_s} + \xi^{\text{ap}}}{1 - \xi^{\text{ap}}} \right) - \left(1 - \frac{\epsilon R_t^H}{1 - \nu R_t^H} \right)} \frac{1}{1 - \nu R_t^H} \stackrel{!}{=} N_{\max}.$$

Note that this approach to calculate N_{\max} assumes the system is stable and has a finite equilibrium value. When the system is out of equilibrium, the value N_{\max} is only an approximation for the number of observed cases at which tracing capacity is overwhelmed.

Reviewer 2

In this manuscript, Contreras et al. evaluate the factors that influence whether ‘test-trace-and-isolate’ (TTI) is sufficient to contain SARS-CoV-2 transmission, even when there is a steady influx of new cases into the population of interest. Control is modeled as isolation of positive cases identified via random testing, syndromic testing and/or contact tracing. They find two tipping points for uncontrolled spread. One is when the reproduction number for the general population remains too high despite preventive measures such as social distancing. The second is when the capacity of the testing and tracing program is exceeded.

Overall, I am very impressed with this manuscript. The writing is very clear and thorough, and the figures convey a lot of information. The motivation for the analyses is laid out well, the methodology strikes a nice balance between simplicity and inclusion of complex transmission dynamics, explicit equations and access to code ensures reproducibility, the results are thoroughly described, and the discussion describes several important implications for public health. Meaningful findings include resource limitations lead to metastable dynamics, TTI can work when preventative measures are in place but not for R of 3.3, and the observed R may underestimate the true R .

We thank the reviewer for their concise summary and the positive evaluation of our manuscript.

However, I would recommend two improvements prior to acceptance into a high-profile journal. The first is to shorten the manuscript as there is a fair amount of redundancy and thus some of the key material may be hard for readers to find.

We agree that our manuscript became quite lengthy. To reduce redundancy, we simplified Figure 4 (resulting in new Fig. 5), and moved the original Figure 4 as well as the accompanying text to the Supplementary Information. We also shortened the part of the discussion that deals with the different testing strategies.

The second is to provide some additional clarifications and reflections on the methods (detailed below). Feedback on methodology: Lines 394/417: There seem to be slightly different definitions of epsilon here. In particular I think there is a distinction between a symptomatic case that does not go into the T pool and a positive contact that doesn’t go into the T pool (since the latter can be asymptomatic)?

We thank the Reviewer for pointing this out; the correct definition is the one of line 417 (line 402 in the current version). We corrected the other occurrence. We assume that the leak term originates from the *new cases* generated in the traced pool T , and not from the current cases in the traced pool. As the leak originates from the *new cases*, a fraction ξ^{ap} of them will be asymptomatic. This allows to write the model in the current form, where the new cases generated by the T

pools is separated by whether they stay in the T pool (proportional to νR_t^H) or they leak into the hidden pool H (proportional to ϵR_t^H). We added a paragraph to the methods section highlighting this (lines 371–375) and modified Figure 1 accordingly. The new paragraph reads:

The traced pool T contains those infected individuals who have been tested positive as well as their positively tested contacts. As these individuals are (imperfectly) isolated, they cause infections with a rate $\nu \Gamma R_t^H$, which are subsequently isolated and therefore stay in the traced pools and additional infections with a rate $\epsilon \Gamma R_t^H$, which are missed and act as an influx to the hidden pools. ν is the isolation factor and ϵ is the leak factor. The overall reproduction number of the traced pool is therefore $R_t^T = (\nu + \epsilon) R_t^H$.

We further added a paragraph to the methods section to highlight the different exchanges between pools (lines 376–381):

In the scope of our model, it is important to differentiate exchanges from pool to pool that are based either on the “reassignment” of individuals or on infections. To the former category belongs the testing and tracing, which transfer cases from the hidden pool to the traced pool. These transfers involve a subtraction and addition of case numbers in the respective pools. To the latter category belongs the recurrent infections ΓR_t^H or $\nu \Gamma R_t^H$ and the ‘leak’ infections $\epsilon \Gamma R_t^H$. Exchanges of this category involves only an addition of case numbers in the respective pool.

Line 395: Strictly speaking it seems that $R = [(1 - \epsilon)\nu + \epsilon]R_t^H$ rather than $(\nu + \epsilon)R_t^H$?

The reviewer is referring to our definition of R_t^T that we introduced in order to represent the number of offspring infections from the traced pool. As discussed in the previous point, individuals in the traced pool can produce two types of offspring infections, (i) those that remain in the traced pool (νR_t^H) and those that leave the traced pool as a leak (ϵR_t^H). In our model, both are independent such that that R_t^T is the sum of both.

Line 458: Can you mention whether you are you assuming that testing yields instantaneous results? Or are resulted delayed, but folks are immediately isolated (seems more relevant for contact tracing)

In our model, testing and isolation happens instantaneously, but reporting happens with a delay, which is modelled as a convolution of the daily new cases and a Gamma kernel (equation 7). We expanded two paragraphs in the methods (model description and testing section) to make this point clearer (lines 350–353 and 448–451). The relevant sentences read:

Model overview. [...] We model the mean-field interactions between the hidden and the traced pool by transition rates which determine the timescales of dynamics of the mode. These transition rates can implicitly incorporate both the time course of the disease and the delays inherent to the TTI process, but we do not explicitly model delays between compartments.

Symptom-driven testing. [...] Testing and isolation happens immediately in this model, but their report into the observed new daily cases \hat{N}^{obs} is delayed. Further real-world delays can effectively be modelled by a lower effective λ_s . In theory, this rate could be increased to one per day. However, this parameter range is on purpose not simulated here.

Line 467: Is there a delay between the time that a case is isolated and when the cases' contacts are isolated? Or is it assumed to be instantaneous?

It is assumed to occur without delay. We added this description to the tracing section (lines 458–461):

Tracing.[...] No delay is assumed here. The parameter η effectively represents the fraction of secondary and tertiary infections that are found through contact-tracing. As this fraction decreases when the delay between testing and contact-tracing increases we assumed a default value of $\eta = 0.66$, i.e. on average only two thirds of subsequent infections are prevented.

Equation 3: It looks like there is an assumption that the proportion of traced H that are Hs and the missed contacts that are Hs are both equal to $(1 - \xi)$. But it strikes me that it might be less than this because symptomatic cases are more likely than asymptomatic cases to be in the T pool?

Regarding the missed contacts in equation 3: these are contacts of isolated individuals which get infected and are missed and not subsequently isolated. As they are new infection, they follow the symptomatic proportion $(1 - \xi)$. Regarding the traced contacts in equation 3: we are assuming that they are removed equally from both asymptomatic and symptomatic pools, because we assume that contacts are equally likely to be found, independently on whether they are

asymptomatic or symptomatic. We therefore can write the equation 3 in this way where the proportion of traced contacts and missed contacts are equal.

Equation 5: It seems that contacts of positive contacts should also be traced (e.g. multiple generations of contact tracing)? Is this reflected in $f(\cdot)$? On the flip side, it seems that some of the contacts might already be in the T pool and that could decrease the value of $f(\cdot)$? Can you comment / clarify?

In order to keep our model simple, we only trace the first generation of cases. This choice biases our model to underestimate the tracing efficiency. As we also assume that individuals are isolated immediately after tracing, which overestimate the tracing efficiency, these two assumptions compensate each other to some degree. Both effects are difficult to assess, which is why we chose this simple description of contact tracing.

Equation 9: Regarding R_{eff} : Since the influx does not represent transmission, I wonder if it should be subtracted from the numerator?

We are interested here in the R as defined by the health authorities, to relate our results to the observations made during the pandemic. In Germany, the health authorities officially publish an R estimate, which is similarly defined as in equation 9 and include the external influx of cases.

Table 2: Do H_a and H_s necessarily reflect non-traced individuals? Or just non-isolated (e.g. can there be non-isolated, traced individuals that are part of H)? Similarly, do T_a and T_s necessarily reflect traced individuals, or just isolated ones?

H_a and H_s necessarily reflect non-traced individuals (in our model we do not consider non-isolated, traced individuals, if someone gets traced, we assume they will be – imperfectly – isolated). Individuals in the T pool (both symptomatic and asymptomatic) are *not necessarily* in isolation because of contact tracing. We address this point amending in the table the definition of the “traced” pool as the *known infected (tested positive) and isolated individuals*.

Minor editing suggestions: Line 32 – ‘surfacing’ – awkward wording

We rephrased the sentence, it now reads (lines 33–35):

Furthermore, SARS-CoV-2 infections generally appear throughout the whole population (not only in regional clusters), which hinders an efficient and quick implementation of TTI strategies.

Lines 321-324: I found these sentences unclear.

We re-wrote the whole section, the relevant sentences now read (lines 290–292 and 299–304):

As the number of available tests is limited, the relative efficiencies of random, symptom-driven and tracing-based testing should determine the allocation of resources [3]. The efficiency of test strategies in terms of positive rate is a primary metric to determine the allocation of tests [4]. [...] Random testing on a population level has the smallest positive rate in the regime of low prevalence that we focus on [5, 4], but could be used in a targeted manner, e.g. screening of health-care workers, highly vulnerable populations [6, 3] or those living in the vicinity of localized outbreaks. We conclude that contact-tracing based testing and highly specific symptoms-based testing should receive the highest priority, with the remaining test capacity used on less specific symptoms-based testing and random screening in particular settings.

Line 397 Consider eliminating ‘quickly waning immunity’ as it is not relevant to the model

We followed the reviewers recommendation and eliminated the phrase.

Line 424: Unclear what is meant by ‘This value causes new infections to be approximately constant’ with $R = 1.8$

In the TTI-stabilized regime, the number of new infections approaches a constant value governed by the external influx and the effectiveness of the TTI strategy – even though the hidden reproduction number R_t^H is greater than one. Therefore, both the effective reproduction number of the whole system and the observed reproduction number are one, making it hard to estimate the hidden reproduction number from case numbers alone. When choosing the parameters of our model, we aimed to both capture the realities of the TTI system and mirror the epidemic situation in Germany during the early summer months, when infections remained approximately constant. Lacking the knowledge of the hidden reproduction number, we assumed $R_t^H = 1.8$, which is on the upper end of the TT-stabilized regime for our default parameter values.

The relevant sentences now read (lines 410–417):

For the default parameters of our model, we used a value of $R_t^H = 1.8$. This parameter was chosen after all others, aiming to mirror the epidemic situation in Germany during the early summer months, when infections remained approximately constant. It is just below the critical value $R_{\text{crit}}^H = 1.98$ for the default scenario, hence $\hat{R}_t^{\text{eff}} = 1$. This value of $R_t^H = 1.8$ is about 54 % lower than the basic reproduction

number $R_0 \approx 3.3$, hence we assume that some non-pharmacological interventions (physical distancing or hygiene measures) are in place, as was the case in Germany during the early summer months [7, 8]. For additional scenarios, we explored the impact of both higher and lower values of R_t^H on our TTI strategy.

Line 457: Would it be more accurate to say λ_s is rate in which symptomatic individuals get tested, amongst the subset who are willing to get tested?

We thank the reviewer for pointing out this inaccuracy, and adopted their definition. It now reads (lines 446–447):

We define λ_s as the daily rate at which symptomatic individuals get tested, among the subset who are willing to get tested.

Line 530: Typo: which section?

We pointed out the right section, the text now reads (lines 528–530):

For instance, following the discussion of Supplementary Section 1, R_{crit}^H was determined by finding the roots of the function returning the real part of the linear system's largest eigenvalue.

Fig 1A/B – Beautiful aesthetics, but hard to understand the details without a figure legend. Also, unclear how this adds value to figure 2

We agree with the reviewer that there is redundancy between figure 1 and figure 2, as our intention was to propose it (figure 1) as graphical abstract. However, as *Nature Communications* does not allow graphical abstracts, we deleted it altogether.

Fig 2B – Lacking legend

We thank the reviewer for pointing this out, and corrected accordingly. The legend now reads:

(b) Simplified version depiction of the model showing the interactions of the two pools. Note that the central epidemiological observables are highlighted in colour: The \hat{N}_t^{obs} (brown) and \hat{R}_t^{obs} (dark red) can be inferred from the traced pool, but the effective reproduction number \hat{R}_t^{eff} (light red) that governs the stability of the whole system remains hidden.

Fig 3 Legend – slightly confusing wording: Does (B,E) correspond to NEW infections only?

Indeed, subfigures B-E present the trends for new daily infections only, both total (N) and observed (\hat{N}^{obs}) new infections.

Fig 5 Hard to see the difference between C and F. Would rescaling help?

The figure was rescaled as suggested.

Figure S2: I like this figure. However, it isn't clear whether decreasing/increasing a parameter ends up decreasing or increasing R_{crit}^H . I wonder if A) would be more informative if each panel was simply a line graph of how the R_{crit}^H changed for particular values of the parameter being considered (rather than show how a distribution of parameters is mapped).

We thank the reviewer for pointing this out. As suggested, we added a new panel a in Supplementary Figure 2 reflecting how R_{crit}^H changed for particular values of the parameter being considered. We also added lines showing the default values of the parameter and R_{crit}^H to provide a reference.

- Table 1: Is $R_0 = 3.3$ used in the model? If not, I'd remove it from the table. Or perhaps it should be listed as a special case of R_t^H ? (And my apologies if I missed where R_0 was included in the model)

Removed, as suggested. R_0 does indeed not play a role, as stated by the reviewer. The idea behind including it in the table was to give a scale for R_t^H , as TTI would not work for the $R_t^H = R_0$ scenario.

Personal observation (no changes needed):

- Lines 159-161. Interesting point. I wonder if you can deduce R_{hidden} , from R_{Obs} and the fraction of cases identified from symptomatic screening vs contact tracing?

We thank the reviewer for pointing this out. It is indeed an interesting point, and one of the follow-up research aims of the group.

Reviewer 3

This was interesting and well-written modelling study on test-trace-isolate strategies to control SARS-COV-2. The model makes a number of simplifying assumptions, especially about contact patterns, but has a nice structure in considering "hidden" and "traced" pools. The model and methods are very clearly laid out and well explained, and the authors should be commended on their sharing of code at the review stage.

We thank the reviewer for their positive evaluation of our manuscript and our open-code philosophy.

The main finding that TTI is going to be most effective when combined with other physical distancing measures is perfectly sensible, and this is supported by other modelling and empirical studies. However, my main concern is that the rather negative message about TTI approaches in places extends beyond the scope of the current model, and is dependent on some key assumptions. To some extent this may be due to the focus in parameters relevant to Germany, but as the messages are intended to be broader it is important that these limitations are very clear.

We added a paragraph in the discussion that puts our results in a larger context and discusses the limitations (lines 323–333):

The parameters of the model have been In other countries, our qualitative conclusions regarding the importance of TTI and the existence of two tipping points will hold, but the parameters would have to be adapted to local circumstances. For instance some Asia-Pacific countries can keep the spread under control, relying to a large extent on test-trace-and-isolate measures after strict and swift initial responses [9]. Factors which contribute to this are (1) significantly larger investment in tracing capacity, (2) a smaller influx of externally acquired infections (especially in the case of New Zealand) and (3) the wider acceptance of mask-wearing and compliance with physical distancing measures in the first place. These countries illustrate that even once "control is lost" in the sense of our model, it can in principle be regained through political measures. A currently discussed mechanism to regain control is the "circuit breaker", a relatively strict lockdown to interrupt infection chains and bring case number down [10]. Such a circuit breaker or reset is particularly effective if it brings the system below the tipping point and thereby enables controlling the spread by TTI again.

Firstly, as far as I can see all of the scenarios assume a constant (and reasonably high) influx of new cases. This is a specific scenario, and one in which TTI is likely to be less effective. When rates of influx into the population are low (e.g. with efficient border control/testing) I think that the sensitivity of the main results to this parameter should be explored in more detail.

We thank the Reviewer for highlighting this issue, and agree that a high constant influx is a special case. In the first part of the manuscript, we assumed that the efficiency of TTI does not depend on the absolute number of cases, and therefore the external influx does not appear in the linear stability analysis. Thus, the critical reproduction number we calculate does not depend on the value of Φ - the first tipping point will be the same even if there is no external influx. In the second part, we assumed that the finite tracing capacity renders TTI less effective when case numbers become too high, consequently the external influx plays a big role here: if it is larger than the tracing capacity, tracing will be overwhelmed quickly. We now added a formal analysis of this parameter in Supplementary Information Sections 2 and 3: For infinite tracing capacity, the external influx only scales the equilibrium number of cases (equation 10). For finite tracing capacity, the number of observed cases at which the capacity is reached depends linearly on the external influx Φ (equation 14).

We now added a sentence to make these two assumptions that separate the first part and the second part of the manuscript clearer (lines 109–112):

Having demonstrated that an effective TTI strategy can in principle control the disease spread, we now turn towards the problem of limited TTI capacity. So far, we assumed that the efficiency of the TTI strategy does not depend on the absolute number of cases. Yet, the amount of contacts that can reliably be traced by health authorities is limited due to the work to be performed by trained personnel: [...]

Secondly, as the authors briefly acknowledge in their discussion, random testing is likely to be most efficient when implemented in specific settings. It is not especially surprising that it doesn't work very well when applied on a population scale. Given this, I think the messaging needs to be very clear about the limited scope of this model for testing the efficacy of random testing in the ways it is likely to be implemented in the real world.

As suggested, we rewrote the entire "testing strategies" subsection in the discussion, and shortened it to highlight the main points (lines 290–304):

As the number of available tests is limited, the relative efficiencies of random, symptom-driven and tracing-based testing should determine the allocation of resources [3]. The efficiency of test strategies in terms of positive rate is a primary metric to determine the allocation

of tests [4]. Contact-tracing based testing will generally be the most efficient use of tests (positive rate on the order of $R_t^H / \{\text{number of contacts}\}$), especially in the regime of low contact numbers [11, 12]. The efficiency of symptoms-driven testing depends on the set of symptoms used for admission: Highly specific symptom sets will allow for a high yield, but misses a number of cases (for instance, 33% of cases don't show a loss of smell/taste [13]). Unspecific symptom sets in contrast will require a high number of tests, especially in seasons where other respiratory conditions are prominent (currently, the fraction of SARS-CoV-2 cases among all influenza-like cases is less than 4 % [14]). Random testing on a population level has the smallest positive rate in the regime of low prevalence that we focus on [5, 4], but could be very useful on a smaller level that is out of the scope of our model, e.g. targeted screening of healthcare workers, highly vulnerable populations [6, 3] or those living in the vicinity of localized outbreaks. We conclude that contact-tracing based testing and highly specific symptoms-based testing should receive the highest priority, with the remaining test capacity used on less specific symptoms-based testing and random screening in particular settings.

Specific comments:

Lines 9-11, this is not how I read the results - suggest expanding a little to say that likely success of TTI is dependent on the reproduction number

We agree that the dependency of the success on the reproduction number was not clearly stated in this sentence, and rephrased the end of the abstract to highlight that TTI alone cannot control an otherwise unhindered spread (lines 10-12):

Our model results suggest that TTI alone is insufficient to contain an otherwise unhindered spread of SARS-CoV-2, implying that complementary measures like social distancing and improved hygiene remain necessary.

Lines 32-33: Not sure this is true (lots of evidence of clustered transmission) - can you elaborate/reference?

We agree that the *transmission* of COVID-19 cases can happen in clusters, but actually wanted to refer to the *socio-geographical distribution* of COVID-19 cases which is more uniformly spread in most countries that have not successfully contained the first wave. We thank the reviewer for highlighting this, and rephrased the sentence accordingly (lines 33-35):

Furthermore, SARS-CoV-2 infections generally appear throughout the whole population (not only in regional clusters), which hinders an efficient and quick implementation of TTI strategies.

Line 39: This might be true for some places, but not others (e.g. NZ) - suggest toning this down/rephrasing

We agree that some countries have successfully curbed the external influx, and rephrased accordingly. The sentence now reads (lines 41–43):

Last, enormous efforts are required to completely prevent influx of COVID-19 cases into a given community, especially during the current global pandemic situation combined with relaxed travel restrictions [15, 16].

Lines 134-135: this seems like a very large instantaneous influx - would a smaller number not be more realistic/sensible to use here?

We thank the reviewer for pointing this out. We would like to remark two points to address this matter: i) the influx Φ accounts for individuals acquiring the infection abroad and then returning to the pools (i.e., they form part of the same total population. e.g., people returning home from vacations) ii) the influx increases the number of hidden cases, it is therefore higher than what would be expected from simply counting those that acquired an infection abroad and were tested after their return.

In fact, we were inspired by the large number of people in Germany that came back from Summer holiday, which led to a transient increase in cases similar to that in our “Default” scenario (July-August). From this perspective, the influx is rather low: during the first two weeks of August, 900 new cases were detected at test centres covering 3 out of 6 highway border Crossing located in the state of Bavaria alone [1]. Given that cars account for only 20% of long-distance holidays [2], one could estimate 4500 observed cases in a single week in August for those crossing Bavarian borders alone - only a lower bound for the number of hidden infections among holidaymakers returning to all of Germany. We now explain this inspiration in the text (lines 117–122):

As an example of how this limited tracing capacity can cause a new tipping point to instability, we simulate here a short but large influx of externally acquired infections (a total of 4000 hidden cases with 92% occurring in the 7 days around $t = 0$, normally distributed with $\sigma = 2$ days, see Fig. 3). This exemplary influx is inspired from the large number of German holidaymakers returning from summer vacation, and is a rather conservative estimate given that there were 900 such

cases observed in the first two weeks of July at Bavarian highway test-centres alone [1].

Additionally, we have explored the impact of a smaller instantaneous influx (Supplementary Figure 3a,b).

Line 171 (and figure 7): While it can be useful to tease apart the contribution of random/symptom-based testing and contact tracing, in reality all three will be used in conjunction. I think perhaps, therefore, that the conclusions about random testing (line 188-191) are a too pessimistic.

We agree that we could have made our focus on random testing at the population level clearer, and rephrased accordingly. The sentence now reads (lines 184–187):

For random testing at the population level to be effective, one would require much higher test rates than currently available in Germany. Random testing nevertheless can be useful to control highly localized outbreaks, and is paramount for screening frontline workers in healthcare, eldercare and education.

Line 266: base reproduction number needs a citation

We thank the reviewer for pointing out this omission, and added a citation (lines 266–268):

Our work, as well as others [17, 18, 19, 20], show that realistic TTI can compensate reproduction numbers of around 1.5-2.5, which is however lower than the basic reproduction number of around 3.3 [21, 22, 23]

Lines 330-340: Suggest a bit of a rewrite here. Random testing is, as you say, likely to be important in specific settings such as hospitals/schools/universities. You should acknowledge that your study can't capture this, and be careful not to come across as though these settings are not important.

We agree that we could have made our focus on random testing at the population level clearer. For brevity, we rewrote the whole part of the discussion that deals with testing strategies. The relevant sentences now read (lines 299–304):

Random testing on a population level has the smallest positive rate in the regime of low prevalence that we focus on [5, 4], but could be used in a targeted manner, e.g. screening of healthcare workers, highly vulnerable populations [6, 3] or those living in the vicinity of localized outbreaks. We conclude that contact-tracing based testing and highly specific symptoms-based testing should receive the highest priority, with the remaining test capacity used on less specific symptoms-based testing and random screening in particular settings.

Lines 405-409: 12% is very much on the low end of rates of asymptomatic individuals. How does increasing this assumption affect your results?

We use 15 % of asymptomatic individuals (The rate are not simply added, but subject to the equation in line 400 and Table 1). We modify this rate in the Fig. S2, where we scan ξ^{ap} (ξ^{ap} depends directly on the rate of asymptomatic individuals). The critical reproduction number changes to some extent, but less than for example under changes of λ_s .

Lines 414-415: this assumption may not be realistic. Do your results hold if you assume that asymptomatic patients are e.g. half as infectious as symptomatics?

We added a supplementary figure (Supplementary Figure 6) which explores this question, and further discussions in supplementary materials (Section 8). The critical reproduction number changes, but qualitatively the results still hold.

References

- [1] David Rising. *Coronavirus positive? Thousands in Germany left wondering*. Washington D.C., USA, Aug. 2020. URL: https://www.washingtonpost.com/world/europe/german-coronavirus-tests-backlog-900-positive-not-yet-told/2020/08/13/c62382e2-dd41-11ea-b4f1-25b762cbbb4%7B%5C_%7Dstory.html.
- [2] Roman Frick and Bente Grimm. *Long-distance Mobility: Current Trends and Future Perspectives*. Tech. rep. Munich: Institute for Mobility Research, 2014. URL: https://www.ifmo.de/files/publications%7B%5C_%7Dcontent/2014/ifmo%7B%5C_%7D2014%7B%5C_%7DLong%7B%5C_%7DDistance%7B%5C_%7DMobility%7B%5C_%7Den.pdf.
- [3] Ezekiel J Emanuel et al. *Fair allocation of scarce medical resources in the time of COVID-19*. 2020.
- [4] Maximilian Kasy and Alexander Teytelboym. “Adaptive targeted infectious disease testing.” In: *Oxford Review of Economic Policy* 36.Supplement_1 (2020), S77–S93. ISSN: 0266-903X. DOI: 10.1093/oxrep/graa018.
- [5] Matthew Cleavelly et al. “A workable strategy for COVID-19 testing: stratified periodic testing rather than universal random testing.” In: *Oxford Review of Economic Policy* 36.Supplement_1 (2020), S14–S37. ISSN: 0266-903X. DOI: 10.1093/oxrep/graa029.
- [6] Lucy Rivett et al. “Screening of healthcare workers for SARS-CoV-2 highlights the role of asymptomatic carriage in COVID-19 transmission.” In: *eLife* 9 (2020), pp. 1–20. ISSN: 2050084X. DOI: 10.7554/eLife.58728.
- [7] Jan Markus Brauner et al. “The effectiveness and perceived burden of nonpharmaceutical interventions against COVID-19 transmission: a modelling study with 41 countries.” In: *medRxiv* (2020).
- [8] Jonas Dehning et al. “Inferring change points in the spread of COVID-19 reveals the effectiveness of interventions.” In: *Science* (2020).
- [9] Emeline Han et al. “Lessons learnt from easing COVID-19 restrictions: an analysis of countries and regions in Asia Pacific and Europe.” In: *The Lancet* 6736.20 (2020), pp. 1–10. ISSN: 1474547X. DOI: 10.1016/S0140-6736(20)32007-9.
- [10] Matt J Keeling et al. “Precautionary breaks planned limited duration circuit breaks to control the prevalence of COVID-19.” In: *medRxiv* (2020), pp. 1–10. URL: <https://medrxiv.org/cgi/content/short/2020.10.13.20211813>.
- [11] Josh A Firth et al. “Combining fine-scale social contact data with epidemic modelling reveals interactions between contact tracing, quarantine, testing and physical distancing for controlling {COVID}-19.” In: *medRxiv* (2020).

-
- [12] Tim Colbourn et al. “Modelling the Health and Economic Impacts of Population-Wide Testing, Contact Tracing and Isolation (PTTI) Strategies for COVID-19 in the UK.” In: *SSRN Electronic Journal* June (2020), pp. 1–80. ISSN: 1556-5068. DOI: 10.2139/ssrn.3627273.
- [13] Alan Chen et al. “Are Gastrointestinal Symptoms Specific for Coronavirus 2019 Infection? A Prospective Case-Control Study From the United States.” In: *Gastroenterology* 0.0 (May 2020). ISSN: 0016-5085, 1528-0012. DOI: 10.1053/j.gastro.2020.05.036. URL: [https://www.gastrojournal.org/article/S0016-5085\(20\)30664-8/abstract](https://www.gastrojournal.org/article/S0016-5085(20)30664-8/abstract).
- [14] Arbeitsgemeinschaft Influenza. *Influenza-Monatsbericht*. https://influenza.rki.de/Wochenberichte/2019_2020/2020-32.pdf. 2020.
- [15] Kevin Linka et al. “Is it safe to lift COVID-19 travel bans? The Newfoundland story.” In: *medRxiv* (2020).
- [16] Nick Warren Ruktanonchai et al. “Assessing the impact of coordinated COVID-19 exit strategies across Europe.” In: *Science* (2020).
- [17] Joel Hellewell et al. “Feasibility of controlling COVID-19 outbreaks by isolation of cases and contacts.” In: *The Lancet Global Health* (2020).
- [18] Luca Ferretti et al. “Quantifying SARS-CoV-2 transmission suggests epidemic control with digital contact tracing.” In: *Science* 368.6491 (2020).
- [19] Emma L Davis et al. “An imperfect tool: COVID-19 ‘test & trace’ success relies on minimising the impact of false negatives and continuation of physical distancing.” In: *medRxiv* (2020).
- [20] Adam J Kucharski et al. “Effectiveness of isolation, testing, contact tracing, and physical distancing on reducing transmission of SARS-CoV-2 in different settings: a mathematical modelling study.” In: *The Lancet Infectious Diseases* 0.0 (June 2020). ISSN: 1473-3099, 1474-4457. DOI: 10.1016/S1473-3099(20)30457-6. URL: [https://www.thelancet.com/journals/laninf/article/PIIS1473-3099\(20\)30457-6/abstract](https://www.thelancet.com/journals/laninf/article/PIIS1473-3099(20)30457-6/abstract).
- [21] Yousef Alimohamadi et al. “Estimate of the Basic Reproduction Number for COVID-19: A Systematic Review and Meta-analysis.” In: *J Prev Med Public Health* 53.3 (2020), pp. 151–157.
- [22] Ying Liu et al. “The reproductive number of COVID-19 is higher compared to SARS coronavirus.” In: *Journal of travel medicine* (2020).
- [23] Ann Barber et al. “The basic reproduction number of SARS-CoV-2: a scoping review of available evidence.” In: *medRxiv* (2020).

REVIEWERS' COMMENTS

Reviewer #1 (Remarks to the Author):

I thank the authors for doing additional analyses in response to my suggestions. I am satisfied with the changes that the authors have made.

Reviewer #2 (Remarks to the Author):

Thank you for your thoughtful consideration of the reviews. I have no further concerns. Congratulations on a job well done!

Seth Blumberg

Reviewer #3 (Remarks to the Author):

The authors have done a thorough job of responding to the reviewer comments. I have no further comments to add, other than that the authors should carefully check their new sections for typos as I spotted a few.

Reviewer 1

I thank the authors for doing additional analyses in response to my suggestions. I am satisfied with the changes that the authors have made.

We thank the reviewer for their positive feedback and all the suggestions.

Reviewer 2

Thank you for your thoughtful consideration of the reviews. I have no further concerns. Congratulations on a job well done!

Seth Blumberg

We thank Dr. Blumberg for the positive feedback and his congratulations.

Reviewer 3

The authors have done a thorough job of responding to the reviewer comments. I have no further comments to add, other than that the authors should carefully check their new sections for typos as I spotted a few.

We thank the reviewer for their positive feedback and all the suggestions. As suggested, we have carefully corrected some minor typos throughout the manuscript.